# MONTAGEAUG: ENHANCING LONG-TAIL ROBUSTNESS AND SEMANTIC CONSISTENCY OF VLMS

## ABSTRACT

Vision-Language Models (VLMs) have made significant strides in multimodal understanding tasks, yet their robustness faces severe challenges when dealing with the long-tail data distributions common in the real world, especially in high-stakes domains like medical image analysis. To address this challenge, we propose MontageAug, a compositional data augmentation approach designed specifically for long-tail vertical domains. It strategically composes images(particularly from head and tail classes) to construct a novel visual scene (a montage image) and synchronously generates a perfectly corresponding compositional text description. This method not only fundamentally guarantees the semantic fidelity of the augmented samples but also effectively alleviates the long-tail data problem by creating information-rich *hard positive samples*. We conducted extensive experimental validation on a model based on the InternVL architecture using ophthalmic medical benchmarks. The results show that MontageAug significantly enhances the model's recognition performance and generalization on tail classes, achieving state-of-the-art (SOTA) performance that surpasses existing augmentation methods on several benchmarks. Furthermore, to explore the approach's extensibility, we validated it on Mathematical Expression Recognition (MER), achieving consistent improvements. Our work ultimately demonstrates that MontageAug, as an efficient, low-cost, and semantics-preserving VLM augmentation strategy, holds practical value in solving the long-tail problem in specialized domains. We plan to open-source our code, benchmark data, and models upon paper acceptance.

## 1 INTRODUCTION

Vision-Language Models (VLMs) have achieved significant breakthroughs in multimodal understanding, demonstrating exceptional capabilities in tasks like assisted diagnosis and report generation (Radford et al., 2021; Liu et al., 2023; Chen et al., 2024c; Li et al., 2023c;b; Chen et al., 2024a; Xu et al., 2025; Bai et al., 2025). However, their real-world generalization is challenged by issues like stylistic data variations (Cai et al., 2024) and the tendency to learn biased *shortcuts* (Brown et al., 2023). Furthermore, in the medical domain, samples of rare diseases (tail classes) are far less common than those of common diseases or normal conditions (head classes). Unlike curated and balanced academic datasets, real ophthalmic datasets collected in clinical practice often exhibit a highly imbalanced *long-tail* distribution of diseases. This can also lead models to learn shortcuts that are overly biased towards head classes, thereby affecting their recognition and understanding of tail classes (Van Horn et al., 2018; Liu et al., 2019).

To more concretely illustrate this challenge, we analyzed the intrinsic distribution of a real clinical ophthalmology dataset FundusGen dataset(Liu & Song, 2025) in practice and its direct impact on model performance, as shown in Figure 1. The figure intuitively presents a typical long-tail phenomenon: a few common conditions, such as diabetic retinopathy, constitute the *head* region with tens of thousands of samples. Meanwhile, the vast majority of other diseases form a *long tail*, with their sample sizes rapidly diminishing, and some rare diseases having fewer than a hundred samples. More critically, the accuracy curve in the figure reveals a problem highly correlated with the data distribution. This is the diagnostic performance we obtained using FundusExpert(Liu & Song, 2025) during an initial hard-negative mining phase. A clear and worrying trend is that the model's performance deteriorates as the data enters the long-tail region. For instance, for diseases with scarce samples like Coloboma and Chorioretinitis, the accuracy is zero. This indicates that

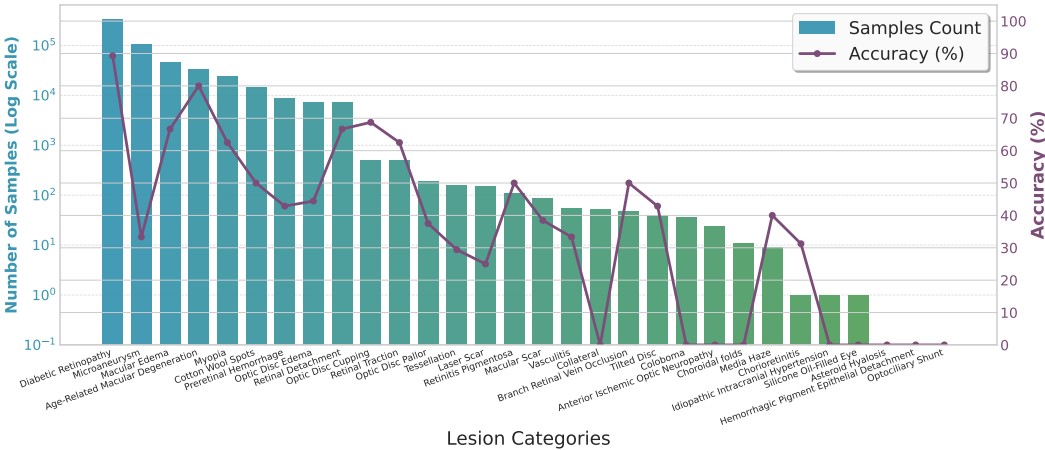

Figure 1: The correlation between the long-tail disease distribution in our authorized FundusGen dataset(Liu & Song, 2025), and the diagnostic accuracy of FundusExpert(Liu & Song, 2025) testing on RFMiD2.0(R_lesion). The green bars (left Y-axis, log scale) represent the number of samples for each disease category. The purple line plot (right Y-axis) shows the accuracy on the corresponding categories, with performance dropping off significantly for tail classes as the sample size decreases.

even state-of-the-art VLMs struggle to escape the constraints of data distribution, tending to learn a biased *shortcut*—overfitting to common diseases while failing to generalize to those that are rare but equally important clinically. This significant performance degradation in the long-tail portion is the core motivation driving this research. The preceding analysis compellingly demonstrates that the long-tail distribution is a key bottleneck limiting VLM performance in real-world clinical environments. A natural approach to solve this problem is to use data augmentation techniques to alleviate data imbalance by expanding the samples of tail classes. However, unlike the mature augmentation techniques in the computer vision domain, data augmentation paradigms specifically designed for VLMs are still underexplored.

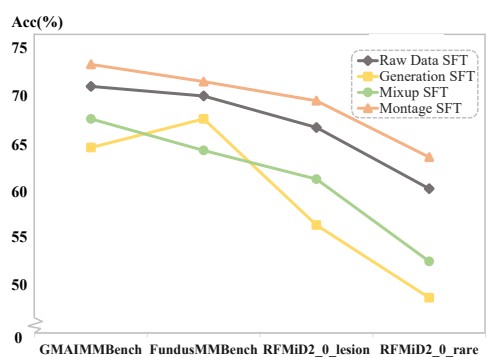

Figure 2: Accuracy comparison of Montage SFT (orange line) with Raw Data SFT, Generation SFT, and Mixup SFT on the fundus images evaluation benchmark. Montage SFT achieves the best performance across all tasks, with a more pronounced advantage on the more challenging tail-class task, RFMID2_0_rare.

Existing mainstream methods struggle to effectively augment scarce tail-class data without causing semantic distortion, primarily due to one core difficulty. This core difficulty lies in the fact that any augmentation for a VLM must strictly adhere to the constraint of image-text semantic consistency. The foundation of a VLM's capability is precise image-text alignment; once an augmentation operation disrupts the correspondence between image content and its textual description, it introduces erroneous supervisory signals, which in turn harms model performance. Consequently, standard augmentation techniques often fall short as they fail to preserve this critical alignment. We highlight the specific limitations of two representative methods below:

**(1) Generative augmentation methods** (Rombach et al., 2022; Ho et al., 2020; Gao et al., 2024) synthesize new image-text pairs to enrich data diversity. However, these methods carry two major risks: the generated images may contain unrealistic artifacts or distort key semantic information, which is unacceptable in high-precision domains like medical diagnosis (Kazeminia et al., 2020; Ning et al., 2025); and high-quality generation often relies on fine-grained text descriptions, which are themselves a scarce and costly resource.

**(2) Mix-based methods** like Mixup (Zhang et al., 2017; Hao et al., 2023) linearly interpolate image pixels, a proven regularization technique in traditional vision tasks. However, this mechanism conflicts with the precise image-text alignment required by VLMs. The resulting non-natural images, which deviate from the real-world data manifold, cause severe semantic attribution problems. For instance, in a Mixup fundus image, key diagnostic features become vaguely overlapped and untraceable, making it difficult for the model to learn precise alignment relationships (Kim et al., 2020; Hao et al., 2023).

The aforementioned bottlenecks indicate that VLM training requires a data augmentation method that can both effectively augment tail data and mechanistically guarantee the absolute consistency of image-text semantics. To this end, we propose the compositional data augmentation method, **MontageAug**. The core idea of MontageAug is to create *hard positive samples*: a sample that is visually novel and complex, but semantically perfectly consistent with its new text. By strategically selecting and combining images from tail classes with images from head classes, MontageAug enriches data diversity and encourages the model to learn the intrinsic features of concepts, rather than simple pattern memorization or over-reliance on specific contexts that appear in a few training samples. We conducted comparative experiments on various VLM architectures (InternVL and LLaVA) and benchmarks in both medical and general domains. The experimental results strongly demonstrate the effectiveness of MontageAug. On the fundus dataset evaluation benchmark (as shown in the Figure 2), our method not only consistently surpasses the standard supervised fine-tuning (Vanilla SFT) baseline but also outperforms other mainstream augmentation methods. These results, along with the results from the LLaVA section in the experiments, indicate that MontageAug, as an efficient, low-cost, and semantics-preserving augmentation strategy, has the potential to significantly enhance VLM performance without introducing any additional data, annotations, or intrusive model modifications. The main contributions of this paper are as follows:

- We propose **MontageAug**, a simple and efficient compositional data augmentation approach for VLMs. It constructs information-rich *hard positive samples* by composing images and their corresponding texts, effectively addressing the long-tail problem while strictly preserving image-text semantic consistency.

- We apply MontageAug to fine-tune a SOTA model based on InternVL that achieves outstanding performance on ophthalmology benchmarks. The model significantly outperforms other methods on the challenging rare disease recognition task, demonstrating its practical value for fundus image analysis.

- Through extension to other vertical domains (Math Expression Recognition), we demonstrate the method's effectiveness in specialized vertical domains.

## 2 RELATED WORK

**Data Augmentation for Vision.**    Data augmentation is a core regularization technique in deep learning, aimed at reducing model overfitting by increasing the diversity of training data (Shorten & Khoshgoftaar, 2019). Early practices focused mainly on geometric transformations (e.g., rotation) and photometric transformations (e.g., adjusting brightness) of images (Krizhevsky et al., 2012). To create more challenging training samples from limited datasets, researchers developed mix-based augmentation techniques such as Mixup (Zhang et al., 2017), CutMix (Yun et al., 2019), and Puzzle Mix (Kim et al., 2020), which involve linear interpolation or regional replacement of images and their labels. However, while these methods have proven extremely effective in single-modality image classification tasks, they do not naturally extend to VLMs, which require enhancing the fine-grained alignment between vision and language. For example, MixGen (Hao et al., 2023), applied to contrastive learning, attempts to establish a fuzzy correlation, but it serves as an effective regularizer for simple tasks like retrieval, not for complex VLM tasks that require precise reasoning and understanding. Another direction for augmenting the input space is to use powerful generative models (Goodfellow et al., 2014; Ho et al., 2020; Rombach et al., 2022) to synthesize entirely new, high-quality image-text pairs. The advantage of generative methods is their ability to create novel combinations of content and style, greatly enriching data diversity. However, this approach may produce unrealistic artifacts, or distort or corrupt key subject information in the images, which is unacceptable in fields with extremely high precision requirements like medical image analysis (Kazeminia et al., 2020; Ning et al., 2025).

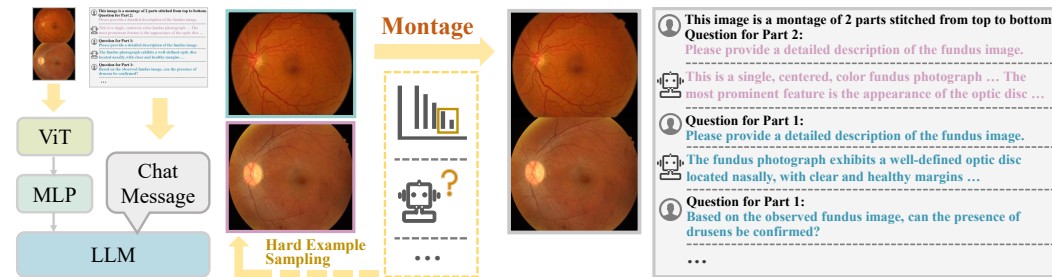

Figure 3: MontageAug Approach for VLM Instruction Fine-tuning.

**Learning from Long-Tail Distributions.** The long-tail distribution is a ubiquitous phenomenon in real-world data, where a few head classes occupy most of the data, while a large number of tail classes have very few samples (Van Horn et al., 2018; Liu et al., 2019). This imbalance causes models to be biased towards head classes during training and perform poorly on tail classes. Traditional methods for addressing the long-tail problem are mainly divided into two categories: data-level re-sampling and algorithm-level re-weighting (Johnson & Khoshgoftaar, 2019). Resampling changes the class distribution of the training set by oversampling the tail classes or undersampling the head classes. Re-weighting assigns different weights to samples from different classes when calculating the loss function, for example, by weighting them by the inverse of the class frequency (Cui et al., 2019). However, simple oversampling can easily lead to overfitting on a few samples, while re-weighting may affect the model's representation learning on head classes and is sensitive to hyperparameters (Cao et al., 2019). MontageAug, by strategically combining images from head and tail classes, not only increases the frequency of appearance of tail classes but, more importantly, it is not a simple repetition. It places tail concepts in new, diverse visual contexts.

## 3 METHOD

The MontageAug data augmentation method derives its name from the *montage* technique in filmmaking. In the film industry, directors can create new narratives and composite emotions that transcend individual shots by splicing together footage from different times and spaces. The core idea of our method is similar: applying the concept of *composition creating new meaning* to the data augmentation of vision and text for VLM training. The entire process primarily consists of three coordinating modules: (1) Hard Sample Prioritization Sampler, (2) Visual Montage Composition, and (3) Textual Montage Composition.

**Hard Sample Prioritization Sampler.** The MontageAug method first constructs a Hard Sample Pool and prioritizes sampling from it. This strategy aims to ensure that the augmentation process focuses more on samples that are more challenging for the model. For instance, in the medical ophthalmology task of this study, we define and screen hard samples from the following two dimensions:

1. Rare samples by category: Directly addressing the long-tail distribution, we classify image-text pairs corresponding to rare ophthalmic diseases as hard samples. This ensures that samples from tail classes receive more adequate exposure during the training process.

2. Difficult samples from model feedback: To further mine challenging samples that lie beyond the model's knowledge boundary, we introduce an automated screening process based on model feedback. First, we use a pre-trained InternVL2.5-8B (Chen et al., 2024b) to process all instruction-tuning data and record its generated answers. Subsequently, we use a powerful Large Language Model (LLM), such as Qwen2.5-72B (Yang et al., 2024), as an automatic evaluator to compare InternVL's outputs with the ground truth (scoring on 1-10 scale, score < 3 identified as hard). Samples where the model's answer is incorrect are identified as difficult samples and added to the hard sample pool.

During training, each batch first loads a main pair. Then, with a preset probability $\alpha$, the MontageAug process is activated. Once activated, the model samples a secondary pair(s) from the constructed hard sample pool for subsequent visual and textual composition. If the hard sample pool is not defined, the method degenerates into random sampling and composition from the entire training dataset.

**Visual Montage Composition.** When the augmentation process is activated, we visually *stitch* the main image $I_{\text{main}}$ with the secondary image(s) $I_{\text{secondary(s)}}$ to create a new montage image $I_{\text{montage}}$. We employ a direct and efficient grid-based composition method, for example, creating a $k \times 1$ or $1 \times k$ grid of images. To ensure a visually natural transition as much as possible, we prefer to select images with similar original edge resolutions for composition and perform appropriate scaling before the operation. This process is akin to editing two separate *shots* together to form a visually richer and more complex *scene*.

**Textual Montage Composition.** As shown in Figure 3, we generate a new compositional text $T_{\text{montage}}$ by concatenating the original text descriptions $T_{\text{main}}$ and $T_{\text{secondary(s)}}$ using a structured template. This generation method is deterministic, has extremely low computational overhead, and preserves the authentic content of the new visual scene. The templates used in this paper to finetune InternVL are as follows:

- **Horizontal:** *"This image is a montage of K parts stitched from left to right. \n Question for Part k: ..."*
- **Vertical:** *"This image is a montage of K parts stitched from top to bottom. \n Question for Part k: ..."*

This rule-based text generation method ensures that the augmented image-text pairs are semantically perfectly consistent, providing an unambiguous training signal for the VLM's alignment learning. Furthermore, this process of constructing *hard positive samples* essentially trains the model's compositional understanding and descriptive abilities, i.e., to accurately *describe what it sees*. The model is required to deconstruct a visually more complex scene and finely align multiple semantic concepts within it with a more structurally complex text. By forcing the model to learn this higher-order correspondence, we can, to some extent, alleviate the problem of hallucination during its reasoning process.

## 4 EXPERIMENT

To verify the generality of the MontageAug method, we conducted instruction fine-tuning experiments on two VLMs with different architectures. The training and evaluation covered both specialized medical domains (fundus diseases diagnosis) and nonmedical vertical domain (mathematical expression recognition).

### 4.1 MEDICAL DOMAIN (FINE-TUNING FOR INTERNVL)

#### 4.1.1 TRAINING DATASETS AND EVALUATION

**Training Datasets.** We utilized the FundusGen dataset (Liu & Song, 2025) (approx. 294K authorized VQA pairs) as the main training data source. We supplemented this with approx. 6K hard samples from MuReD(Rodríguez et al., 2022) and RFMiD 2.0(Panchal et al., 2023) training sets. For these supplementary images, we used Gemini 2.5 Pro to generate high-quality medical reports based on the image, reliable and detailed structured labels. More data details are shown in Appendix B.1. Finally, we screened for hard samples according to the hard-sample-prioritized target sampling method mentioned in Section 3. The overall dataset has a ratio of normal samples to hard samples of approximately 11:4.

**Evaluation Datasets.**

GMAIMMBench (Ye et al., 2024). GMAIMMBench is a broad medical evaluation benchmark. We selected the fundus images subset from GMAIMMBench for evaluation, and subsequent mentions of GMAIMMBench in this paper refer to this subset (abbreviated as **GMAI** for table display). It broadly covers diagnostic tasks for dozens of fundus diseases, with a total of 312 entries.

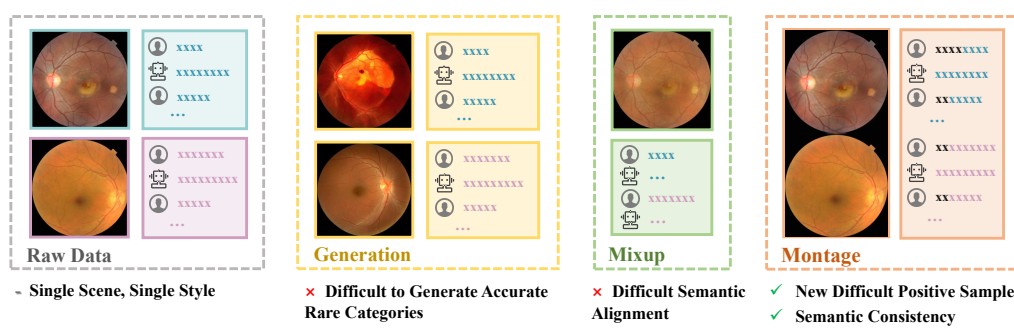

Figure 4: Different Data Augmentation Applied to VLMs.

FundusMMBench (Liu & Song, 2025). FundusMMBench is a benchmark specifically designed for fundus images evaluation (abbreviated as **Fundus**), containing 620 data items for the recognition, diagnosis, and symptom grading of more common fundus diseases.

RFMID2.0 (Panchal et al., 2023). The RFMiD 2.0 dataset is a multi-label classification dataset of fundus images, providing annotations for 45 eye diseases. We selected subsets of samples with diseases from the test and validation sets of RFMID2.0 (abbreviated as **R_lesion**) and samples of rare diseases (abbreviated as **R_rare**). Using the original disease labels, we constructed ten-option multiple-choice questions suitable for VLM evaluation, with one correct option and nine distractors, using simple prompts like "View the fundus image provided. What is the condition demonstrated?". RFMiD2_0_lesion has 293 data items, and RFMiD2_0_rare has 183 data items. More details are shown in Appendix B.2. These datasets cover the evaluation of various fundus diseases, and we report the accuracy on each dataset. We use weak matching for evaluation (referencing VLMEvalKit(Duan et al., 2024) settings, details shown in Appendix D.1). For cases where the model does not fully follow the instructions, we use GPT-4o for answer extraction and matching.

### 4.1.2 PERFORMANCE COMPARISON

To validate the effectiveness of MontageAug, we compared it with a series of representative baseline methods, as well as with state-of-the-art models in the field and closed-source commercial models. The instruction fine-tuning experiments in this section were all conducted on 8 NVIDIA A100 GPUs, fine-tuning the parameters of ViT, MLP, and LLM of InternVL-Chat-V3.0(Zhu et al., 2025). We used the same training hyperparameters for all methods (lr=2e-5, bs=128, epoch=2, etc.) to ensure a fair comparison.

**Vanilla SFT** : Standard supervised fine-tuning without any specific augmentation or sampling strategy, serving as the baseline reference for the effectiveness of our method.

**Hard Samples Oversampling**: An idea derived from classic long-tail learning methods. It balances the training data distribution by oversampling tail classes. We estimated the frequency at which hard samples were *additionally* exposed in the MontageAug experiments and then converted this frequency into a direct oversampling rate (repeat_time) for these datasets. The core principle of the comparison is to ensure that the total number of times the model encounters hard sample categories is roughly equal between this setting and the MontageAug setting, to verify that the advantage of our method is not solely due to data balancing.

**Generate Hard Samples** (Johnson & Khoshgoftaar, 2019): A method to directly enrich the diversity of tail data. For the hard samples used in the MontageAug method, we additionally invoked the advanced fundus generation model RetinaLogos (Ning et al., 2025) to synthesize new images (with the text descriptions unchanged) to augment the training set. This aims to contrast with MontageAug's *compositional* paradigm of enriching image content features. In this method, the VLM extracts descriptive content from the text labels to be used as input for RetinaLogos to generate training samples. On 8 NVIDIA A100 GPUs with a batch size of 16, RetinaLogos required an additional 40 hours for inference to generate the necessary hard samples.

Table 1: Accuracy of different methods on fundus images benchmarks. Bold values mean best performance.

| Method | GMAI(%) | Fundus(%) | R_lesion(%) | R_rare(%) | Average(%) |
|---|---|---|---|---|---|
| InternVL3.0-8B | 42.3 | 46.5 | 25.0 | 18.0 | 33.0 |
| Vanilla SFT (Baseline) | 70.8 | 69.8 | 66.5 | 60.1 | 66.8 |
| Oversampling | 71.5(+0.7) | 69.5(-0.3) | 67.6(+1.1) | 60.1(+0.0) | 67.2(+0.4) |
| Generation | 64.4(-6.4) | 67.4(-2.4) | 56.3(-10.2) | 48.6(-11.5) | 59.2(-7.6) |
| Mixup | 67.4(-3.4) | 64.1(-5.7) | 61.1(-5.4) | 52.5(-7.6) | 61.3(-5.5) |
| Expand_bs | 68.9(-1.9) | 70.5(+0.7) | 66.9(+0.4) | 59.6(-0.5) | 66.5(-0.3) |
| **MontageAug** | **73.1(+2.3)** | **71.3(+1.5)** | **69.3(+2.8)** | **63.4(+3.3)** | **69.3(+2.5)** |
| OphthaReason-Intern | 37.8 | 40.8 | 24.9 | 11.5 | 28.8 |
| FundusExpert | 66.7 | 69.7 | 45.4 | 32.3 | 53.5 |
| GPT4o | 41.6 | 57.4 | 25.3 | 30.1 | 38.6 |
| Gemini-2.5-Pro | 65.4 | 47.4 | 29.0 | 31.1 | 43.2 |

**Mixup** (Zhang et al., 2017; Hao et al., 2023): A representative baseline for mix-based sample augmentation. When applied to VLMs, we select images of similar resolution, resize them to the same dimensions, and then superimpose them pixel-wise in a 1:1 ratio, while retaining all text dialogue labels to simulate its application in a multimodal scenario. The augmentation activation probability $\alpha$, decay parameter and the number of superimposed samples $k$ remain the same as MontageAug. A hard-sample-prioritized sampling strategy was also used.

**Effective Batch Size Expansion**: To investigate whether the performance improvement of MontageAug arises simply from a larger "effective" batch size (i.e., processing more visual concepts per step via montage), we conducted this experiment. In the setting (**Expand_bs**), the original images that would have formed a montage were instead added to the same training batch as independent samples. This ensures the total information exposure per step is identical to MontageAug.

**MontageAug**: The augmentation activation probability $\alpha$ was set to 0.4, with a decay parameter of 0.6 (linearly decaying the activation probability after 60% of training), and the number of composed samples $k$ was set to 2.

Furthermore, to assess the competitiveness of our approach against established standards, we compare it with the following state-of-the-art specialized models and commercial systems:

**Domain-Specific SOTA Models**: **FundusExpert (8B)** (Liu & Song, 2025) is a VLM specialized for fundus images using a positioning-diagnosis collaboration approach. **OphthaReason-Intern (2B)** (Wu et al., 2025) is a multimodal reasoning model based on reinforcement learning (RL) designed specifically for the ophthalmology domain.

**Commercial Models**: **GPT-4o** and **Gemini-2.5-Pro** are included to demonstrate the performance of representative closed-source commercial models on this task.

The results are shown in Table 1, from which we can draw the following key conclusions:

**1. MontageAug significantly outperforms standard fine-tuning and traditional long-tail strategies.** Compared to the **Vanilla SFT** baseline, MontageAug achieves significant performance improvements across all benchmarks, with a particularly pronounced advantage on the **RFMiD2.0_rare** dataset, which is specifically designed to evaluate the recognition of rare diseases. More valuably, MontageAug shows stronger performance compared to methods that increase tail data by direct duplication (**Hard Samples Oversampling**) or synthesizing new samples (**Generate Hard Samples**). This compellingly demonstrates that the advantage of our method does not simply stem from increasing the exposure of tail data, but from its core mechanism—forcing the model to learn more generalizable compositional features by constructing information-rich *hard positive samples*. It is noteworthy that simple oversampling showed no performance improvement on the rare disease dataset RFMiD2.0_rare and led to a slight performance drop on Fundus, possibly reflecting its inherent risk of overfitting (Cao et al., 2019).

**2. Semantic fidelity is key for VLM data augmentation.** The comparison with **Mixup** highlights the importance of semantic consistency. Mixup, through pixel-wise linear interpolation, generates

Table 2: Ablation on different training settings in MontageAug on fundus images benchmarks (accuracy). Bold values mean best performance.

| SFT Method | Settings | Parameters | GMAI | Fundus | R_lesion | R_rare |
|---|---|---|---|---|---|---|
| No Augmentation | Ablation of training epochs | 1e | 66.7% | 66.2% | 64.8% | 59.6% |
| | | 2e | 66.3% | 65.3% | 65.5% | 59.6% |
| MontageAug | Ablation of training epochs | 1e, $\alpha = 0.4$, $k = 2$ | 67.0% | 66.5% | 65.5% | 60.1% |
| | | 2e, $\alpha = 0.4$, $k = 2$ | **67.6%** | **67.3%** | 67.2% | **61.2%** |
| | | 3e, $\alpha = 0.4$, $k = 2$ | 67.0% | 66.8% | **67.6%** | 61.2% |
| | Ablation of probability $\alpha$ | 2e, $\alpha = 0.2$, $k = 2$ | 67.0% | 66.0% | 66.6% | 60.1% |
| | | 2e, $\alpha = 0.6$, $k = 2$ | 66.3% | 65.5% | 65.2% | 58.5% |
| | Ablation of strategies | 2e, Random Sampling | 66.7% | 66.5% | 65.9% | 59.6% |
| | Ablation of complexity $k$ | 2e, $\alpha = 0.4$, $k = 3$ | 65.1% | 65.8% | 65.5% | 60.1% |

samples that are visually unnatural and severely misaligned with their text descriptions, leading to poor performance. In contrast, MontageAug ensures complete semantic alignment of the augmented image-text pairs by synchronously composing both the visual and textual elements, providing high-quality supervisory signals to the model. This comparison clearly indicates that augmentation that sacrifice semantic fidelity are not advisable for VLM that rely on precise image-text understanding.

**3. MontageAug demonstrates strong domain competitiveness.** In comparison with state-of-the-art specialized models in the domain (FundusExpert, OphthaReason-Intern) and top-tier closed-source commercial models (GPT-4o, Gemini-2.5-Pro), the general-purpose InternVL model fine-tuned with MontageAug exhibits highly competitive performance. This proves that MontageAug is an efficient and economical strategy that can rapidly adapt and enhance the capabilities of powerful foundational VLMs in specialized domains (such as medical image analysis) without requiring intrusive modifications to the model architecture.

**4. Improvements derive from contextual composition, not data throughput.** Simply expanding the effective batch size (Expand_bs) to match MontageAug's information throughput resulted in an average accuracy of 66.5%, which is significantly lower than MontageAug (69.3%). Notably, performance on the rare disease subset (R_rare) dropped to 59.6%, whereas MontageAug achieved 63.4%. This result rules out the hypothesis that the gains are solely due to increased effective batch size, confirming that the *visual composition* of hard positive samples creates a unique and necessary learning context that forces the model to perform deeper visual reasoning.

### 4.1.3 ABLATION STUDY

The dataset for the ablation study is a randomly sampled subset (20%) of the full fundus images data used in the previous experiments. The instruction fine-tuning experiments in this section were all conducted on 4 NVIDIA A100 GPUs, fine-tuning the parameters of ViT, MLP, and LLM. We used the same learning rate (lr=2e-5) and batch size (bs=32) for all methods.

**Ablation on training epochs.** (Table 2) From 1 to 2 epochs, model without augmentation performance did not consistently improve, while the MontageAug model showed a significant performance improvement. Further increasing the epochs to 3 (MontageAug), we observed that the overall improvement plateaued relative to the increased computational cost. Thus, we maintain the **2-epoch** setting as the optimal trade-off between computational efficiency and performance for this task.

**Ablation on data augmentation probability** $\alpha$**.** (Table 2) The final scheme selects an optimal MontageAug enhancement probability of 0.4. When $\alpha$ is 0.2, the potential of the MontageAug method is not fully realized, whereas an $\alpha$ of 0.6 leads to a performance drop, constrained by the misalignment between high-frequency MontageAug training and evaluation.

**Ablation on target sampling strategy.** (Table 2) We compared the *hard sample prioritization* sampling strategy with random sampling. The results show that the version using tail-prioritized sampling outperformed random sampling by about 2 percentage points overall. This indicates that targeted selection of data from hard samples can effectively improve performance.

Table 3: Ablation on textual scaffolding in MontageAug design on fundus images benchmarks(acc).

| Method | GMAI | Fundus | R_lesion | R_rare | Average |
|---|---|---|---|---|---|
| Vanilla SFT | 66.3% | 65.3% | 65.5% | 59.6% | 64.2% |
| **MontageAug(full)** | **67.6%** | **67.3%** | **67.2%** | **61.2%** | **65.8%** |
| w/o Textual Scaffolding | 56.7% | 59.2% | 58.0% | 53.0% | 56.7% |
| Image Montage Only | 64.1% | 64.0% | 62.8% | 54.4% | 61.3% |
| w/o Meta-instruction | 65.4% | 64.8% | 64.5% | 57.4% | 63.0% |

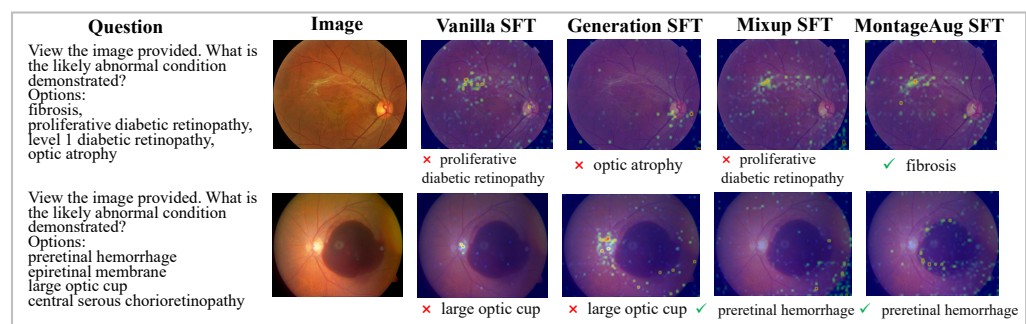

Figure 5: Comparison of attention maps on rare disease samples. `MontageAug` model can accurately focus its attention on pathological features and make correct judgments.

**Ablation on MontageAug complexity k.** (Table 2) We explored combining different numbers of images in a single Montage image and ultimately chose a combination of k=2 for this task.

**Ablation on Textual Scaffolding.** (Table 3) We conducted an ablation study on textual scaffolding in MontageAug design under the k=2 setting. `w/o Textual Scaffolding` means no templates are added to the text; instead, the two original text descriptions $T_{main}$ and $T_{secondary(s)}$ are simply concatenated. `Image Montage Only` means randomly keeping the text annotation corresponding to one image and the corresponding connecting text. `w/o Meta-instruction` means keeping the block structure like Part k, but removing the introductory sentence at the beginning (i.e., "This image is a montage of..."). As shown in Table 3, `w/o Textual Scaffolding` caused a significant performance drop to 56.7%, emphasizing the critical role of precise image-text alignment. Furthermore, `Image Montage Only` or `w/o Meta-instruction` both performed worse than the complete MontageAug (65.8%), confirming the necessity of the holistic design of *Vision + Text + Structured Instruction.*

### 4.1.4 VISUALIZATION ANALYSIS

To more intuitively understand how `MontageAug` improves model performance, especially in enhancing disease diagnosis, we visualized the model's attention maps during inference. We selected a series of samples and compared the attention distributions.

As shown in Figure 5, the visualization results reveal behavioral differences in models under different training strategies. When asked about the condition in the image, models like `Vanilla SFT` failed to focus on key lesion areas or, while focusing on problematic areas, could not accurately identify them. For instance, in judging "preretinal hemorrhage", the `Generation SFT` model ignored the diagnostically valuable lesion area, and while the `Mixup SFT` model answered correctly, it did not truly understand the pathological features. In contrast, the `MontageAug SFT` model exhibited more focused attention, with its focal point aligning with the lesion areas that ophthalmologists would focus on. This indicates that by introducing complex visual scenes composed of *hard positive samples* during training, `MontageAug` successfully forces the model to move beyond simple memorization of superficial statistical features of the data and instead learn more robust and generalizable deep visual representations of the diseases themselves.

Table 4: Performance Comparison on UniMER Validation Set (Metrics: BLEU ↑ / Edit Distance ↓). Hard subsets are marked. Bold values mean best performance.

| Method | SPE | CPE (Hard) | SCE | HWE (Hard) |
|---|---|---|---|---|
| | BLEU↑ / Edit↓ | BLEU↑ / Edit↓ | BLEU↑ / Edit↓ | BLEU↑ / Edit↓ |
| UniMERNet-Small | 0.911 / 0.064 | 0.917 / 0.065 | 0.569 / 0.243 | 0.891 / 0.075 |
| **+ MontageAug** | **0.915 / 0.061** | **0.934 / 0.053** | **0.577 / 0.241** | **0.904 / 0.067** |

Table 5: Performance Comparison on Public Handwritten Datasets (Metric: ExpRate ↑).

| Method | CROHME 2014 | CROHME 2016 | CROHME 2019 | HME100K |
|---|---|---|---|---|
| Baseline | 59.43% | 57.63% | 55.88% | 65.08% |
| **+ MontageAug** | **62.17% (+2.7%)** | **61.81% (+4.2%)** | **58.63% (+2.8%)** | **66.57% (+1.5%)** |

## 4.2 EXTENSIBILITY TO NONMEDICAL DOMAINS: MATHEMATICAL EXPRESSION RECOGNITION (MER)

To demonstrate that MontageAug is not limited to ophthalmology and addresses the general problem of long-tail distributions in vertical domains, we extended our validation to Mathematical Expression Recognition (MER). This task faces severe long-tail distribution problems, particularly with rare, complex, or long formulas.

**Task:** The dataset is categorized into four types: Simple Print Expression (SPE), Complex Print Expression (CPE), Screen Capture Expression (SCE), and Handwritten Expression (HWE). Among these, SPE constitutes the *Head* classes, while CPE and HWE are considered *Tail/Hard* classes due to their structural complexity and scarcity.

**Settings:** We applied MontageAug to the SOTA model UniMERNet (Wang et al., 2024). We trained on the UniMER dataset (1.06M samples) and evaluated on the UniMER Validation set and public benchmarks (CROHME, HME100K). MontageAug synthesizes longer, more complex formula samples by randomly splicing two formula images. Since MER is a direct content parsing task that does not require explicit VQA prompting, we adapted the textual composition strategy: for horizontally spliced images, the original LaTeX labels are simply concatenated with a space; for vertically spliced images, they are joined by a newline character ('\n'). All other experimental hyperparameters, including the augmentation probability ($\alpha = 0.4$) and decay strategy, remain consistent with the default MontageAug settings described in Section 4.

**Results on UniMER Validation Set:** As shown in Table 4, MontageAug consistently outperforms the baseline. Crucially, the gains are most significant on the long-tail/hard subsets (CPE and HWE), boosting BLEU scores by 1.7% and 1.3% respectively.

**Results on Public Benchmarks:** We further evaluated on CROHME 2014/2016/2019 (Mouchere et al., 2014) and HME100K. As shown in Table 5, MontageAug achieved new SOTA results by significantly improving Expression Accuracy (ExpRate), with a 2.7% gain on CROHME 2014.

## 5 CONCLUSION

In this paper, we focused on the severe challenges that Vision-Language Models (VLMs) face when processing the long-tail distributions common in the real world, especially in high-stakes domains like medical image analysis. To address this issue, we proposed `MontageAug`, a compositional data augmentation approach that enhances model robustness by constructing *semantically-faithful hard positive samples*. Extensive experiments on multiple ophthalmology benchmarks demonstrated that `MontageAug` can significantly improve the model's ability to recognize rare categories without introducing additional computational overhead at inference time. Its performance surpassed various baseline methods, achieving state-of-the-art results. Our extended experiments on Mathematical Expression Recognition further verified the method's generalizability in diverse vertical domains. In conclusion, `MontageAug` provides a simple, efficient, and low-cost new path toward building more robust VLMs.

## 6 ETHICS STATEMENT

All authors have read and agree to abide by the ICLR Code of Ethics. The authors declare no conflicts of interest. The medical data utilized in this study consists of publicly available datasets and in-house retrospective data. The collection and use of the in-house data have been fully authorized and were conducted in compliance with all relevant ethical regulations and guidelines.

## 7 REPRODUCIBILITY STATEMENT

To ensure the reproducibility of our research, we provide the following details. Code: The code used for our experiments, along with the benchmark data and trained models, will be made publicly available on a GitHub repository upon acceptance of this paper. Our study utilizes several publicly available datasets for training, all of which are cited in Appendix B.1. The in-house training data cannot be released due to patient privacy and institutional data sharing policies. Implementation Details: All experiments were conducted using the InternVL-Chat-V3.0 and LLaVA-v1.5-7B models. The training was performed on NVIDIA A100 GPUs. Detailed hyperparameters and experimental settings for all models and methods are provided in the experimental sections (Section 4 and Appendix E).

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

## A  LLM DISCLAIMER

During the preparation of this manuscript, we utilized a large language model (LLM) to assist in refining and improving the fluency and coherence of the text in the Introduction, Related Work, and Method sections. The LLM was also used for assistance with LaTeX formatting. The core scientific contributions, including the proposed methods, experimental design, and analysis of results, are entirely the original work of the authors.

## B  FUNDUS DATA SOURCE

### B.1  TRAINING DATA SOURCE

**Main Data (approx. 97%):**  We obtained authorized use of the FundusGen dataset(Liu & Song, 2025). This part contains about 294K VQA pairs. FundusGen is a high-quality dataset built specifically for collaborative reasoning in fundus images, which already contains rich VQA pairs.

**Supplementary Data (approx. 3%):**  We additionally constructed about 6K data samples from the open-source training sets of MuReD(Rodríguez et al., 2022) and RFMiD 2.0(Panchal et al., 2023). Regarding report generation for the supplementary images, we used Gemini 2.5 Pro(Comanici et al., 2025) as a powerful MLLM to generate medical reports. The template (prompt) we used is shown below; it instructs the model to act as a professional ophthalmologist and generate objective fundus image descriptions based on structured labels (labels_json) and authoritative disease descriptions (disease_descriptions from the RFMiD 2.0 paper(Panchal et al., 2023)).

---

**Prompt to Generate Reports**

```
You are a professional fundus imaging specialist. You have a color
fundus image of a patient, along with detailed corresponding labels
:

{labels_json}

{disease_descriptions}

Combined with this information, please provide a detailed
description of this fundus image. This description may reference
findings that may aid in the diagnosis, but should not directly
cite or infer a diagnosis. Please note that only objective
descriptions of fundus images are required.
```

---

### B.2  LESION CATEGORIES AND RARE CATEGORIES IN RFMiD2.0

**Lesion Categories.**  This primary group encompasses a wide spectrum of 40 retinal findings. It includes common, high-impact diseases such as Age-Related Macular Degeneration and Diabetic Retinopathy, as well as specific pathological signs like Cotton Wool Spots and Microaneurysms. This comprehensive set is designed to test the model's overall diagnostic accuracy across a diverse range of conditions. The full list includes:

Age-Related Macular Degeneration, Anterior Ischemic Optic Neuropathy, Asteroid Hyalosis, Branch Retinal Vein Occlusion, Central retinal vein occlusion, Chorioretinitis, Choroidal folds, Collateral, Coloboma, Cotton Wool Spots, Diabetic Retinopathy, Drusens, Epiretinal Membrane, Haemorrhagic Retinopathy, Hemorrhagic Pigment Epithelial Detachment, Idiopathic Intracranial Hypertension, Laser Scar, Macular Edema, Macular Hole, Macular Scar, Media Haze, Microaneurysm, Myopia, Optic Disc Cupping, Optic Disc Edema, Optic Disc Pallor, Optic Neuritis, Optociliary Shunt, Preretinal Hemorrhage, Retinal Detachment, Retinal Traction, Retinal pigment epithelium changes, Retinal tears, Retinitis, Retinitis Pigmentosa, Silicone Oil-Filled Eye, Tessellation, Tilted Disc, Tortuous Vessels, Vasculitis.

Figure 6: Schematic diagram of MontageAug composition methods.

---

**Algorithm 1** MontageAug Training

---

**Require:** training batch $B$, augmentation probability $\alpha$, montage complexity $k$
1: **for** $(I_{\text{main}}, T_{\text{main}})$ in $B$ **do**
2:     **if** random() $< \alpha$ **then**
3:         $(I_{\text{sec}}, T_{\text{sec}}) \leftarrow$ sample_from_hard_pool$(k-1)$         ▷ 1. Hard Sample Prioritization
4:         $I_{\text{montage}} \leftarrow$ visual_montage$(I_{\text{main}}, I_{\text{sec}}, k)$         ▷ 2. Visual Montage Composition
5:         $T_{\text{montage}} \leftarrow$ textual_montage$(T_{\text{main}}, T_{\text{sec}}, k)$         ▷ 3. Textual Montage Composition
6:         loss $\leftarrow$ compute_loss(model($I_{\text{montage}}, T_{\text{montage}}$))
7:     **else**
8:         loss $\leftarrow$ compute_loss(model($I_{\text{main}}, T_{\text{main}}$))
9:     **end if**
10:    loss.backward()
11:    optimizer.step()
12: **end for**

---

**Rare Categories.** This group is a curated subset of 30 lesion categories identified as being less frequently encountered. It is specifically designed to challenge the model's ability to recognize and classify rare pathologies, providing a focused assessment of its performance on conditions with limited data prevalence. The categories are:

Anterior Ischemic Optic Neuropathy, Asteroid Hyalosis, Chorioretinitis, Choroidal folds, Collateral, Coloboma, Drusens, Epiretinal Membrane, Haemorrhagic Retinopathy, Hemorrhagic Pigment Epithelial Detachment, Idiopathic Intracranial Hypertension, Laser Scar, Macular Edema, Macular Hole, Macular Scar, Media Haze, Optic Disc Pallor, Optic Neuritis, Optociliary Shunt, Preretinal Hemorrhage, Retinal Traction, Retinal pigment epithelium changes, Retinal tears, Retinitis, Retinitis Pigmentosa, Silicone Oil-Filled Eye, Tessellation, Tilted Disc, Tortuous Vessels, Vasculitis.

## C METHOD DETAILS

### C.1 MONTAGE COMPOSITION DIAGRAMS

When MontageAug is activated, we visually *compose* the main image $I_{\text{main}}$ with the secondary image(s) $I_{\text{secondary(s)}}$ to create a new montage image $I_{\text{montage}}$. We employ a direct and efficient grid-based composition method, for example, creating a $k \times 1$ or $1 \times k$ grid of images. To ensure a visually natural transition as much as possible, we prefer to select images with similar original edge resolutions for composition and perform appropriate scaling before the operation.

We summarize the training steps of MontageAug in Algorithm 1.

### C.2 CONSTRUCTION OF THE HARD-SAMPLE POOL

We used an LLM(Qwen2.5-72B) to combine the question and reference answer to score the answer provided by InternVL2.5-8B(Chen et al., 2024b) on a scale of 1-10. Samples with a score lower than 3 entered the hard sample pool. It should be noted that the threshold of 3 was not an arbitrarily chosen hyperparameter tuned to improve performance. It is a quality-based binary distinction. Scores lower than 3 (specifically, 1–2) correspond to cases with severe hallucinations or major errors. In contrast, scores of 3 or higher are interpreted as demonstrating at least partial correctness.

**Consistency and Robustness Analysis**: To address the issue of dependency on a specific evaluator, we conducted a consistency analysis. We randomly sampled 500 samples for validation (using the same prompt and threshold $< 3$). The Intersection over Union (IoU) of the hard sample pool between Qwen2.5 and GPT-4o was 91.2% (125/137), and the IoU with Gemini 2.5 Pro was 90.7% (127/140). This indicates that the choice of a specific model has a minor impact on the composition of the hard sample pool. The following is the prompt used.

---

**Prompt to Score the Answer**

```
### System Prompt:
You are an expert AI evaluator.
Your task is to evaluate the quality of a candidate answer
generated by a Vision-Language Model (VLM) regarding a fundus image
, based on a given Reference Ground Truth.

### Input Data:
1. Question: {}
2. Reference Answer: {}
3. Candidate Answer: {}

### Scoring Rubric (1-10 Scale):
Please assign a score based on the following strict criteria:

- Score 1-2 (Critical Failure / Hallucination):
  - The answer provides a wrong diagnosis
  (e.g., identifying normal fundus as diseased).
  - Contains severe hallucinations (describing lesions that do not
exist).
  - Completely contradicts the Reference.

- Score 3-4 (Partially Correct / Inaccurate Details):
  - The core diagnosis is arguably correct or vague, BUT the answer
contains minor hallucinations or incorrect descriptive details
  (e.g., wrong location/color of lesions).
  - Misses important pathological evidence.

- Score 5-6 (Acceptable / Incomplete):
  - No factual errors or hallucinations.
  - The description lacks significant detail
  compared to the Reference.

- Score 7-8 (Good / Minor Omissions):
  - Accurate diagnosis with a largely complete description.
  - Only minor details are missing.
  - Fluent and clinically sound.

- Score 9-10 (Excellent / Professional):
  - Perfectly aligns with the Reference in diagnosis and detail.
  - Comprehensive, precise terminology.

### Output Format:

Return the result in JSON format strictly:
{
  "reasoning": "Brief explanation of the evaluation...",
  "score": <integer_between_1_and_10>
}
```

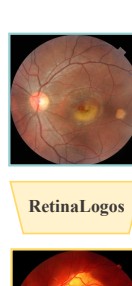
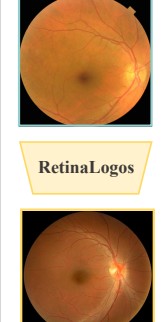

This is a well-centered, color fundus photograph of the posterior pole. The image is of good clarity and illumination, allowing for a detailed assessment of the optic disc, retinal vasculature, and macula. Optic Disc: The optic disc is located nasally and presents with distinct, well-demarcated margins. The neuroretinal rim appears pink and healthy. The lamina cribrosa is not visible. The major retinal vessels emerge centrally from the disc in a normal configuration. Retinal Vasculature: The retinal arterioles exhibit an increased light reflex and appear somewhat attenuated, resulting in a reduced artery-to-vein ratio of approximately 1:3. The vessels follow a normal branching pattern without significant tortuosity. There is no evidence of neovascularization, hemorrhages, microaneurysms, or vascular occlusions. Macula and Posterior Pole: The normal foveal reflex is absent. Centered within the macula is a large, prominent, roughly circular subretinal lesion. This lesion is approximately 1.5 disc diameters in size and demonstrates a complex appearance. It consists of a dense, elevated ring of yellowish-white material surrounding a central area with mixed reddish-orange and hyperpigmented discoloration. The retinal surface overlying and surrounding this lesion appears uneven. Temporal to this large lesion, a smaller, discrete, oval-shaped yellowish deposit with well-defined borders is also noted. Retinal Background and Periphery: The surrounding retinal pigment epithelium has a mottled appearance. There are no other focal lesions, exudates, hemorrhages, or tears visible within the field of view. The vitreous appears clear.

This is a high-quality, well-illuminated color fundus photograph of a single eye. The media is clear, allowing for a sharp view of the retinal structures. Optic Disc: The optic disc is located to the right of the macula and presents with a round shape and distinct, well-defined margins. The neuroretinal rim has a healthy orange-pink hue and appears robust and uniform in its entire circumference. The central physiologic cup is small. There is no evidence of disc edema, pallor, or peripapillary hemorrhage. Retinal Vasculature: The retinal blood vessels emerge from the optic disc and follow a normal branching pattern across the fundus. The arteries are a lighter red and narrower than the veins, which are a darker red and wider, consistent with a normal artery-to-vein ratio. The vessels have a normal, gentle course without significant tortuosity. There are no signs of arteriovenous nicking, sheathing, emboli, or neovascularization. Macula and Fovea: The macula, located temporal to the optic disc, appears flat and has a normal, slightly darker pigmentation compared to the surrounding retina. A distinct foveal reflex is present at the center of the macula, indicating normal foveal contour. There are no hemorrhages, exudates, drusen, or other pigmentary abnormalities in the macular region. Retinal Background and Periphery: The retinal background displays a uniform orange-red coloration throughout. The underlying retinal pigment epithelium appears homogenous. No hemorrhages, cotton wool spots, laser scars, or other retinal lesions are visible in the posterior pole or the observable periphery.

Figure 7: RetinaLogos generates images.

## C.3 Fundus image generation method

For the hard samples used in the MontageAug method, we additionally invoked the advanced fundus generation model RetinaLogos (Ning et al., 2025) to synthesize new images (with the text descriptions unchanged) to augment the training set. This aims to contrast with MontageAug's *compositional* paradigm of enriching image content features, as shown in Figure 7. In this method, the VLM extracts descriptive content from the text labels to be used as input for RetinaLogos to generate training samples. On 8 NVIDIA A100 GPUs with a batch size of 16, RetinaLogos required an additional 40 hours for inference to generate the necessary hard samples.

## D Experiment

### D.1 Evaluation Prompt Details

We followed VLMEvalKit(Duan et al., 2024) settings for weak matching. The prompt used is: Our evaluation process on fundus color images referenced the custom MCQ (multiple choice) dataset settings of VLMEvalKit(Duan et al., 2024).

We append "Answer with the option's letter from the given choices directly." to the tested question, requiring the model to reply only with the option. The post-processing flow follows: Rule Matching → LLM Judgment. Statistics show that less than 1% of samples require LLM judgment.

Regarding evaluator variance: for the inference results in Table 1 of the main text, when we switched the post-processing evaluator to Gemini 2.5 Pro(Comanici et al., 2025) or Qwen2.5-72B(Yang et al., 2024), the final accuracy remained consistent.

The post-processing prompt is as follows:

**Prompt to Post-processing**

```
You are an AI assistant who will help me to match an answer with
several options of a single-choice question.
You are provided with a question, several options, and an answer,
and you need to find which option is most similar to the answer.
If the meaning of all options are significantly different from the
answer, output Z.
Your should output a single uppercase character in A, B, C, D (if
they are valid options), and Z.

Example 1:
Question: What is the main object in image?
```

```
Options: A. teddy bear B. rabbit C. cat D. dog
Answer: a cute teddy bear
Your output: A

Example 2:
Question: What is the main object in image?
Options: A. teddy bear B. rabbit C. cat D. dog
Answer: Spider
Your output: Z

Example 3:
Question: {}?
Options: {}
Answer: {}
Your output:
```

## D.2 COMPUTATIONAL EFFICIENCY ANALYSIS

To evaluate the computational overhead of different augmentation methods in practical applications, we recorded the time taken for each method in the same hardware environment (8 NVIDIA A100 GPUs). The results are shown in Table 6. The "Preprocessing Time" in the table refers to the one-time, offline computational cost required before formal training begins, such as hard sample screening or image synthesis for generative methods.

**Definition of Epoch and Iterations** In our experimental setup, 1 Epoch is defined as one complete traversal of the original dataset ($N$). It is important to distinguish that `MontageAug` operates as an online augmentation strategy (maintaining the same number of iterations per epoch as standard SFT), whereas `Oversampling` involves a physical expansion of the dataset, resulting in an increased number of iterations per epoch ($N + N_{hard}$).

Table 6: Computational efficiency and performance comparison on fundus images benchmarks (The performance results are from Table 1).

| Method | Preprocessing (h) | Training (h) | Total (h) | Average Perf. |
|---|---|---|---|---|
| Vanilla SFT | - | 38.2 | 38.2 | 66.8% |
| Oversampling | 6.0 | 47.2 | 53.2 | 67.2% |
| Generation | 6.0 + 40.6 | 47.2 | 93.8 | 59.2% |
| Mixup | 6.0 | 41.5 | 47.5 | 61.3% |
| **MontageAug (Ours)** | 6.0 | 38.3 | 44.3 | **69.3%** |

**Performance vs. Cost** From the Table 6, it is clear that the core training time of `MontageAug` (38.3 hours) is almost identical to that of the baseline `Vanilla SFT` (38.2 hours). This fully demonstrates that the online image composition and text combination operations performed by our method during the training loop have negligible computational overhead, ensuring extremely high training efficiency. `MontageAug` shares the same hard sample screening preprocessing cost (6 hours) with `Hard Samples Oversampling`. However, because the oversampling method increases the frequency of tail samples by duplicating data, its training time (47.2 hours) is about 23% longer than our method. What's more, online `Mixup` operations make its training time about 8% longer than our method. Finally, `Generation` is the most computationally expensive (93.8 hours total) due to the significant time required for image synthesis.

In summary, among all data augmentation methods that yield performance improvements, `MontageAug` demonstrates the highest computational efficiency. It achieves the best performance gain with a one-time, controllable preprocessing investment, with almost no increase in training time and no extra resource consumption during inference, proving its value as an efficient, practical, and easy-to-deploy augmentation strategy.

### D.2.1 Detailed Analysis of Effective Augmentation Rate

We further clarify the relationship between the nominal augmentation probability parameter $\alpha$ and *actual effective augmentation rate*. Actual effective augmentation rate is lower than the nominal rate (take 0.4 as an example), due to two specific implementation details designed to ensure training stability and image quality:

- **Probability Decay Strategy:** The augmentation probability $\alpha$ is not constant throughout the training. It is maintained at 0.4 for the first 60% of the training epochs and then linearly decays to 0 over the remaining 40%. This results in an average probability of:

$$\bar{\alpha} = 0.6 \times 0.4 + 0.4 \times \left(\frac{0.4 + 0}{2}\right) = 0.24 + 0.08 = 0.32 \tag{1}$$

- **Quality Constraints & Failure Rate:** To strictly prevent image distortion, our implementation performs rigorous aspect ratio similarity checks (restricting the resolution ratio at the connection interface to $< 1.2$). If a resolution-matching image cannot be found in the hard sample pool within a limited number of attempts (set to 50), the augmentation is skipped for that instance. Monitoring of the training process in Table 1 indicates a skip rate of approximately 20% under these constraints.

Therefore, the *actual effective augmentation rate $P_{eff}$* is calculated as:

$$P_{eff} = \bar{\alpha} \times (1 - \text{Failure Rate}) = 0.32 \times (1 - 0.2) = 0.256 \tag{2}$$

### D.3 Supplementary Visualization Results

To qualitatively analyze the model's reasoning process and validate that MontageAug enhances focus on relevant pathological features, we implemented a method to visualize the model's internal attention mechanisms during inference. This methodology allows us to generate attention heatmaps that highlight the image regions the model deems most important when making a diagnostic prediction.

As shown in Figure 8, the visualization results reveal behavioral differences in models under different training strategies. When asked about the condition in the image, models like `Vanilla SFT` failed to focus on key lesion areas or, while focusing on problematic areas, could not accurately identify them. For instance, in judging "preretinal hemorrhage", the `Generation SFT` model ignored the diagnostically valuable lesion area, and while the `Mixup SFT` model answered correctly, it did not truly understand the pathological features. In contrast, the `MontageAug SFT` model exhibited more focused attention, with its focal point aligning with the lesion areas that ophthalmologists would focus on. This indicates that by introducing complex visual scenes composed of *hard positive samples* during training, `MontageAug` successfully forces the model to move beyond simple memorization of superficial statistical features of the data and instead learn more robust and generalizable deep visual representations of the diseases themselves.

## E  Generality and Architecture Dependency Analysis on LLaVA

To assess the general potential of `MontageAug` and its adaptability to different model architectures, we applied it to the widely used open-source VLM, LLaVA-1.5(Liu et al., 2024a). This appendix provides the detailed experimental setup, full results, and analysis. LLaVA-v1.5-7B is a widely used and easily accessible open-source VLM, used to evaluate its performance on general domain tasks. We used the standard setting of LLaVA-1.5. The core motivation was to establish a fair, reproducible baseline to purely assess the performance boundaries of the data augmentation strategy itself.

### E.1 Experimental Setup

**MontageAug Settings:** It is noteworthy that this exploratory experiment employed a simplified strategy: utilizing only the core image composition mechanism of `MontageAug` with random

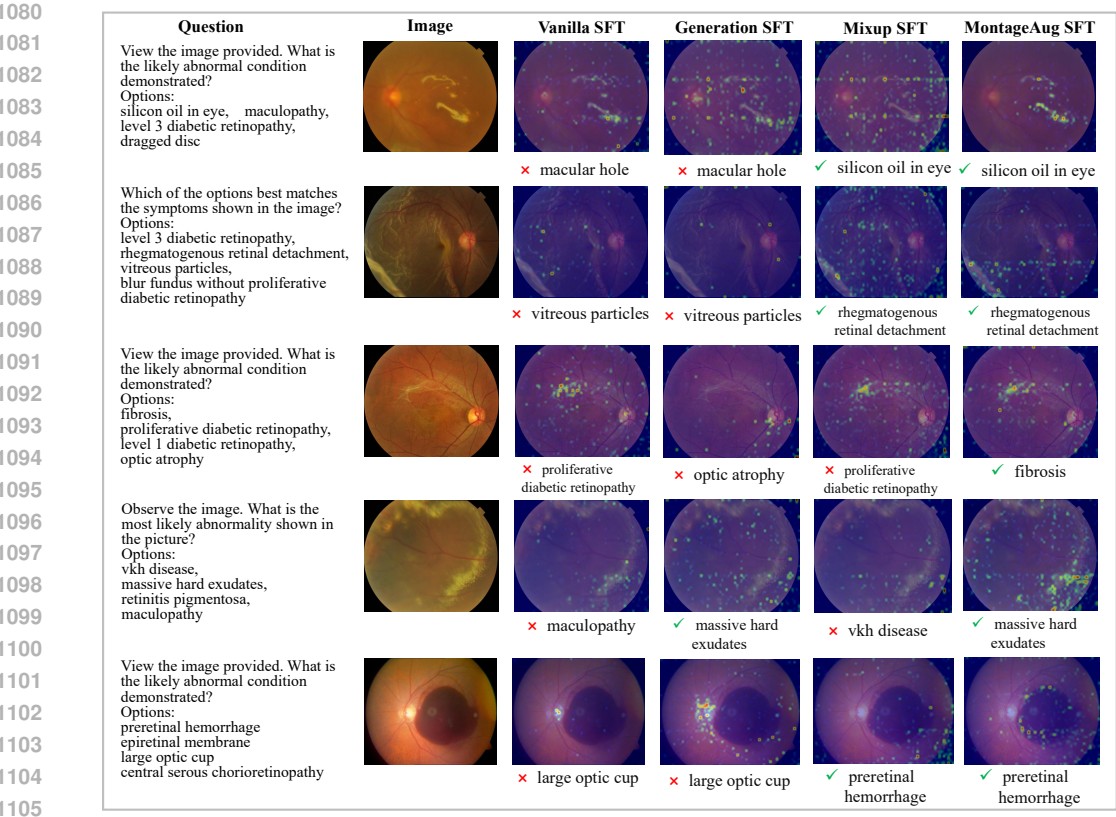

Figure 8: Comparison of attention maps on rare disease samples. The model trained with `MontageAug` can accurately focus its attention on pathological features and make correct judgments.

sampling, without activating the *hard sample prioritization* strategy used in our medical domain experiments. This was done to isolate and examine the universality of `MontageAug`'s core compositional idea and its interaction with the model architecture.

**Training and Evaluation:** We used the officially open-sourced 665k instruction fine-tuning dataset from LLaVA (Liu et al., 2024a) for training. To ensure a fair comparison, the hyperparameter configurations for this experiment, including learning rate, batch size, number of epochs, and the fixed resolution of 336x336, strictly followed the settings published in the original LLaVA-1.5 paper (Liu et al., 2024a). **Evaluation Benchmarks:** To comprehensively evaluate the model's performance, we selected a series of recognized general-purpose VLM benchmarks, including MMBench (Liu et al., 2024b), SEED-Bench (Li et al., 2023a), TextVQA (Singh et al., 2019), GQA (Hudson & Manning, 2019), VQAv2 (Goyal et al., 2017), SciQA-IMG (Lu et al., 2022), POPE (Li et al., 2023d), MME (Fu et al., 2023), and VisWiz (Gurari et al., 2018).

### E.2 FULL RESULTS AND ANALYSIS

#### E.2.1 PERFORMANCE COMPARISON AND ANALYSIS

The results, as shown in Table 7, where the Vanilla SFT results are directly copied from the original paper, reveal a clear pattern of differentiation: even under this simplified application, `MontageAug` still brings significant performance improvements on benchmarks that test complex compositional reasoning abilities. For instance, on the **MM-Vet** benchmark, which is designed to evaluate comprehensive capabilities, performance improved by **2.4** points, and on the science question-answering benchmark **SciQA-IMG**, a significant gain of **3.1** points was achieved. We attribute this to the fact that the montage process creates information-rich *hard positive samples*, which implicitly train the

Table 7: Extensibility experiment results of MontageAug on LLaVA-1.5.

| Benchmark | Vanilla SFT | + MontageAug | Change ($\Delta$) |
|---|---|---|---|
| SciQA-IMG | 66.8 | 69.9 | +3.1 |
| MM-Vet | 31.1 | 33.5 | +2.4 |
| SEED-Bench (img) | 66.1 | 67.2 | +1.1 |
| GQA | 62.0 | 62.5 | +0.5 |
| VQAv2 | 78.5 | 79.0 | +0.5 |
| TextVQA | 58.2 | 58.4 | +0.2 |
| MMBench (en) | 64.3 | 64.3 | 0.0 |
| POPE | 85.9 | 85.8 | -0.1 |
| VisWiz | 50.0 | 49.6 | -0.4 |
| MME | 1510.7 | 1502.6 | -8.1 |

model's core abilities to deconstruct complex visual scenes, understand compositional concepts, and integrate diverse visual-language information—skills that are directly evaluated by these benchmarks.

In contrast, on benchmarks that heavily rely on fine-grained visual perception, the model's performance showed stagnation or a slight decline, especially on **MME** (-8.1) and **TextVQA** (+0.2%, no significant change). We attribute this performance divergence to a key interaction between our data augmentation strategy and LLaVA-1.5's inherent *fixed-resolution architecture*. LLaVA-1.5 forces all visual inputs into a fixed 336x336 resolution. Consequently, larger compositional montage images must be downsampled, which inevitably leads to a *significant loss of high-frequency visual details*. This degradation in visual fidelity directly impairs the model's ability to perform tasks that depend on perceptual acuity, such as reading tiny characters for OCR in TextVQA or performing fine-grained object recognition and counting in MME.

Furthermore, composing semantically unrelated images (like an ocean and a book) also increases the risk of the model hallucinating when faced with simple polling tasks (e.g., the POPE dataset asks if something is in the picture, where the composed image might cause confusion).

For convenience, we directly applied the MontageAug augmentation method to LLaVA, but in practice, specific models require certain modifications (e.g., supporting variable or higher resolutions and employing more intelligent sampling strategies).

The experimental observations reveal the deep relationship that data augmentation cannot exist independently of model architecture. Our findings may provide an suggestion for future research: efficient multi-modal data augmentation strategies (like MontageAug) should be *Data-Model Co-designed* to fully release their potential.

