# OpenReview forum: "MontageAug: Enhancing Long-tail Robustness And Semantic Consistency of VLMs"
_ICLR.cc/2026/Conference — Submitted to ICLR 2026_

### Official Review · Reviewer_xo91 · 2025-10-24

**Soundness:** 2
**Presentation:** 1
**Contribution:** 2
**Rating:** 4
**Confidence:** 4

**Summary:**

This paper proposes MontageAug, an augmentation method to enhance VLMs' long-tail robustness. MontageAug comprises three major components: (1) Hard Sample Prioritization Sampler, which prioritizes the sampling of rare and hard samples; (2) Visual Montage Composition, which vertically concatenates images from head and tail classes to form an augmentation image; (3) Textual Montage Composition, which concatenates captions through a template to ensure semantic consistency. Experimental results show that MontageAug surpasses simple oversampling and single-modality-based augmentation (mixup and caption-based image generation) in medical and general-purpose VLM benchmarks.

**Strengths:**

1. It is sound to consider semantic consistency in long-tail augmentation methods for VLM, instead of single-modality augmentation.
2. The experimental results show that MontageAug surpasses simple oversampling and single-modality augmentation.

**Weaknesses:**

1. **Over-claimed generalizability.**  The paper has claimed its effectiveness in the general domain. However, the experimental results on general-purpose benchmarks are only left in the appendix, with only one baseline (simple SFT), which is not convincing.

2. **Confusing writing.** Some important details are missing, such as training data and MLLM for report generation. And the organization of the paper is also confusing. For detailed questions, please see below.

3. If the evaluation benchmarks are from the medical domain, why does this paper choose a general-purpose VLM as the base model instead of widely-used medical VLMs like LLaVA-Med?

**Questions:**

- Line 265: Which powerful MLLM is used for report generation?
- Sec 4.1.1, what open-source data has been used?
- In Figure 3, how does the model learn to differentiate part 1 and part 2? My concerns stem from the experimental results on general benchmarks that MontageAug suppresses performance on tasks that rely on fine-grained perception (e.g., MME). Does this really relate to resolution, or due to the misalignment between different parts?
- Line 130, it is suggested to replace "comprehensively surpasses" with "consistently surpasses"
- Missing '.' in the captions of Figure 3 and Figure 4.

---

> ### Author Response · Authors · 2025-11-21
>
> Dear Reviewer xo91,
>
> We thank you for recognizing our method for considering semantic consistency in long-tail augmentation methods for VLMs and for affirming that our experimental results surpass baselines.
>
>
> ### Response to W1 (Generalizability Claim)
>
> We sincerely thank the reviewer for this insightful comment.
>
> 1. In the revision, we explicitly narrow the scope of our contribution. Instead of positioning MontageAug as a universal augmentation method, we strictly define it as a specialized solution for **Long-Tail Vertical Domains** (e.g., specialized medical imaging, specialized OCR), where data scarcity and class imbalance are bottlenecks. To robustly support our revised claim (effectiveness in vertical long-tail domains), we have added new cross-domain experiments (please see **General Response** for details).
>
> 2. Explanation of general domain results.
>
> - Due to space limitations, we prioritize presenting results from vertical domains in the main text that directly reflect our main contributions.
>
> - Baseline Selection & Reproducibility: LLaVA-1.5 is currently the most standard and easily reproducible open-source baseline in the community. We utilized standard SFT as the primary baseline to ensure reproducibility within the community's standard training recipes.
>
> - Lack of Comparable Methods: Unlike in vision tasks, "plug-and-play" data augmentation methods specifically designed for VLMs that preserve semantic consistency are still underexplored, limiting available comparative baselines.
>
> - Inapplicability of Naive Sampling:  In massive general-domain datasets, defining and extracting "tail" concepts without specific prior analysis is non-trivial compared to structured vertical datasets.
>
> ---
>
> ### Response to W2, Q1, and Q2 (Data and Models)
>
> Thank you for the rigorous suggestion. We clarify the data composition and report generation models as follows:
>
> * **Main Body (~294K entries):** Comes from the authorized **FundusGen dataset** [1]. According to the original paper, this data was primarily generated by **GPT-4o**, and it already contains high-quality reports/VQA pairs.
> * **Supplementary Data (~6K entries):** Sourced from **MuReD** and **RFMiD 2.0**. The "powerful MLLM" we used for report generation here is **Gemini 2.5 Pro**. From the results in Table 1, Gemini 2.5 Pro demonstrates strong basic capabilities in this field. The template (prompt) we used is shown below; it instructs the model to act as a professional ophthalmologist and generate objective fundus image feature descriptions based on the image itself, reliable and detailed structured labels (`labels_json`), and authoritative disease descriptions (`disease_descriptions` from the RFMiD 2.0 paper [2]). We have explicitly stated this in Section 4.1.1 of the revised PDF and added the following prompt to appendix B.1 (line 779).
>
> > f"You are a professional fundus imaging specialist. You have a color fundus image of a patient, along with detailed corresponding labels: {labels_json}.\n\n"
> > f"{disease_descriptions}\n\n"
> > "Combined with this information, please provide a detailed description of this fundus image. This description may reference findings that may aid in the diagnosis, but should not directly cite or infer a diagnosis. Please note that only objective descriptions of fundus images are required."
>
> **Correction regarding Appendix B.1:**
> We realize that the presentation of the source of training data in Appendix B.1 of the original manuscript may have been somewhat misleading. The numerous datasets listed in the original Appendix B.1 (e.g., IDRID, BRSET, In-house Data, etc.) are actually the data sources mentioned in the FundusGen paper[1]; the collection, organization, and annotation of this data were completed by the authors of FundusGen.
>
> We have revised Appendix B.1 removing the original list extracted from the FundusGen paper and have directly stated the dataset composition as:
> 1.  FundusGen dataset (authorized)
> 2.  MuReD (Training Set)
> 3.  RFMiD 2.0 (Training Set)
>
> *[1] Xinyao Liu and Diping Song. Constructing ophthalmic mllm for positioning-diagnosis collaboration through clinical cognitive chain reasoning. ICCV 2025.*

---

> ### Author Response · Authors · 2025-11-21
>
> ### Response to W3 (Base Model Choice)
>
> Thank you for your suggestion. We chose InternVL over LLaVA-Med for the following reasons:
>
> 1.  **Match with Augmentation Strategy:** As a general-purpose model supporting **high-resolution input** (its original architecture possesses higher visual processing capabilities), InternVL is better matched to our MontageAug, which is an "information-intensive" augmentation strategy. In contrast, LLaVA-Med inherits the fixed low resolution of LLaVA-1.5, limiting the effectiveness of image splicing.
> 2.  **Performance Capability:** Choosing InternVL, a model with stronger general capabilities and higher visual processing power, allows us to maximize the demonstration of the true potential of the MontageAug method in solving long-tail problems.
>
> Additionally, we supplemented experimental results analyzing base model capabilities. As shown in Table R7, **InternVL3.0-8B** outperforms LLaVA-Med in both overall and tail-class (R_rare) performance. This indicates that InternVL has a higher initial capability starting point for our specific medical tasks. Furthermore, after SFT, InternVL's final average performance reached **66.8%**, far higher than LLaVA-Med's **56.0%**.
>
> **Table R7: Base Model Comparison (LLaVA-Med vs InternVL)**
>
> | Method | GMAI(%) | Fundus(%) | R_lesion(%) | R_rare(%) | Average(%) |
> | :--- | :---: | :---: | :---: | :---: | :---: |
> | LLaVA-Med (7B) | 34.0 | 26.4 | 14.7 | 12.0 | 21.8 |
> | LLaVA-Med_sft | 57.7 | 53.4 | 59.0 | 53.6 | 56.0 |
> | InternVL3.0-8B | 42.3 | 46.5 | 25.0 | 18.0 | 33.0 |
> | **InternVL3.0-8B_sft** | **70.8** | **69.8** | **66.5** | **60.1** | **66.8** |
>
> ---
>
> ### Response to Q3 (Differentiation and Alignment)
>
> Thank you for your question. We confirm that the model differentiates regions via explicit instructions, and the performance drop on some fine-grained perception tasks is attributed to resolution bottlenecks rather than alignment issues.
>
> **1. How the model differentiates Part 1 and Part 2:**
> We used explicit structured instructions (e.g., *"This image is a montage... Question for Part 1:..."*). This mandatory "Vision-Text" alignment forces the model to evolve sparse and precise attention mechanisms. During the SFT phase, the model not only learns spatial mapping but also learns to actively suppress visual interference from the other part, focusing attention on the specific region.
>
> **2. Main cause of performance drop is resolution limitation, not Misalignment:**
> LLaVA-1.5's input resolution is fixed at $336 \times 336$. The Montage operation reduces the effective resolution of a single image, causing a loss of high-frequency details. This is confirmed by two aspects:
>
> * **No Logic Confusion:** If the model confused different Parts (Misalignment), strong logic reasoning tasks (**SciQA-IMG**, **MM-Vet**) should have collapsed first. However, they actually improved by **3.1%** and **2.4%** respectively, proving the model did not get confused and possesses stronger logical reasoning capabilities.
> * **Pixel Loss in Fine-Grained Tasks:** Performance drops were concentrated in **MME** and **VisWiz**. The VisWiz dataset is taken by blind users, where original image quality is often poor (blur, poor lighting) and often requires recognizing tiny objects or text (e.g., medicine bottle labels). MME also contains massive Fine-grained Perception subtasks like recognizing small text. During training, LLaVA further compresses images under fixed resolution, making visual details in Montage samples unrecognizable. Such a training process may have "sacrificed" the model's sensitivity to tiny visual features in exchange for stronger global semantic integration capabilities.
>
> ---
>
> ### Response to Q4 and Q5 (Writing Issues)
>
> Thank you for the corrections. We changed "comprehensively surpasses" to the more precise **"consistently surpasses"** (line 126) and complete the missing punctuation in the captions of Figure 3 and Figure 4 in the revised PDF.

---

### Official Review · Reviewer_WyrV · 2025-10-30

**Soundness:** 1
**Presentation:** 3
**Contribution:** 2
**Rating:** 4
**Confidence:** 4

**Summary:**

The authors present a novel augmentation technique called MontageAug specifically created for VLMs and long-tailed problems, which conserves the semantic coherence of image-text pairs by leveraging “hard-positive” samples.
MontageAug creates a montage of multiple images and a text template to pair it with the respective textual descriptions. Further, the method chooses images based on the class abundance and on the instance difficulty, which is assessed by the performance of a base model. MontageAug achieves SOTA performance on Ophthalmology datasets outperforming simple SFT as well as other augmentation approaches like generative augmentations, sampling techniques or mixing techniques.

**Strengths:**

MontageAug convinces with its simplicity and broad applicability to most VLMs and tasks.
The method is well motivated in the need of improving models for long-tailed problems.

**Weaknesses:**

1.) It is unclear how an epoch is defined for each method and therefore it is unclear whether the presented performance gain is solely due to the fact that MontageAug sees more image-text pairs e.g. compared to SFT. While Table 2 indicates an overfitting with more epochs for SFT (i.e. 2 epochs), it remains to be shown if sampling based methods would benefit from more epochs here. Sampling techniques could benefit from more epochs in this case, e.g. increasing the numbers of samples seen to the amount of image-text pairs the MontageAug sees.

2.) Further, due to the montage the "effective" batch size could be larger in the case of MontageAug which could be the source of improvement, an ablation with SFT and Oversampling with a 40% (since the best $\alpha$ for MontageAug was 0.4) increased batch size would be interesting as well. To be more explict: one could conduct an experiment where instead of the montaged images those images which would make up the montage will end up in the same batch with a batch size accomodating the additional images of the montage so that the information exposure per training step is the same.

3.) Due to the fact that the epoch for each method is not properly defined, the training times presented in table 3 are against expectation i.e. why the training time for “Oversampling” should be higher than the “MontageAug”/”Vanilla SFT” method.

4.) It is unclear (cf lines 265ff) how the training dataset was generated. Both the used MLLM/VLM and the used templates are not specified. Further, to be more clear the authors should already link to the listing of the training datasets within the appendix around lines 265.

5.) Regarding the textual montage the exact template being used is not provided.

6.) The captions of tables 1, 2 and 4 could be improved as they do not mention the metrics being used as well as what bold values are supposed to mean (i.e. best perfromance). In table 3 it could be specified which "medical dataset" was being used (e.g. referring to the main body).

7.) While there is a small hyper-parameter grid-search done for the MontageAug method the other ablated methods were only done with one set of possible sub-optimal hyper-parameters which could make MontageAug superior due to a better set of hyper-parameters with respect to the other methods. E.g. for the hard oversampling one could find a more suitable $\alpha$ (oversampling frequency).

8.) Line 144: The authors state that their method demonstrates practical value for medical image analysis in general while it is only shown for Ophthalmology. I.e. this claim is too broad for what is shown in the paper.

9.) Line 48ff: How many private samples were collected? Should also be included in the caption of Figure 1.

10.) Line 270ff: In listing their validation data they don't state the number of classes for all datasets (i.e. GMAIMMBench and FundusMMBench).

11.) Line 411: The authors state "... the MontageAug method shows continuous performance improvement with more training epochs" whereas it is only shown for 1 and 2 epochs so the statement is too general.

12.) Why did the authors chose LLaVA-1.5 to ablate the method on other tasks, as it is clear that the method benefits strongly from a VLM which supports dynamic resolution? Overall section 4.2 feels rushed.

13.) The authors should explicitly refer to figure 1 in line 130.

14.) Regarding the reading flow the listing starting from line 095 feels abrupt and a little out of context. Further in line 337 there should be some transition between the listing of the comparison methods and the baseline methods.

15.) The statement in line 030/031 and 144 is too broad "demonstrating its practical value for medical image analysis.", while it is only shown for ophthalmology images.

**Questions:**

How exactly are epochs defined in your paper? If an epoch is one run through all training cases how can it be that in Table 3 the training time of Vanilla SFT and Oversampling differ so much? If the Oversampling technique additionally sees 40% more rare cases on top than one would expect a training time 40% larger (around 53h) than SFT which is not the case.

Why is the increase in training time of the MontageAug method only marginal compared to SFT in table 3, if InternVL uses dynamic resolution there should be 40% more tokens being processed?

---

> ### Author Response · Authors · 2025-11-21
>
> Dear Reviewer WyrV,
>
> We thank you for recognizing MontageAug's simplicity, broad applicability, and the soundness of its motivation regarding long-tail problems. Your questions are professional and in-depth, particularly regarding the training mechanics and experimental fairness. Here are our detailed responses.
>
> ### Response to W1, W2, W3, and Q1 (Regarding Epoch, Batch Size, and Efficiency)
>
> We thank the reviewer for the insightful questions regarding the definition of training Epochs and the fairness of comparison. Below, we clarify the definitions, explain the differences in training time, and present new ablation experiment results regarding "effective batch size."
>
> **1. Definition of Epochs:**
> In our method, **1 Epoch is defined as one traversal of the original dataset ($N$)**. MontageAug is an online augmentation (same number of iterations as SFT), whereas Oversampling is a physical expansion of the dataset (iterations = $N + N_{hard}$).
>
> **2. Why Oversampling time is lower than expected:**
> The reviewer correctly pointed out that if we simply added 40% more data, the training time for Oversampling should theoretically increase by about 40%, yet the time we reported is lower. This discrepancy arises because the **actual effective augmentation rate is lower than the nominal 0.4**, due to two specific implementation details designed in our code to ensure training stability and image quality:
>
> * **Probability Decay:** As described in Section 4.1.2, $\alpha$ is not constant. It remains 0.4 for the first 60% of the training process, then decays linearly to 0. The average $\alpha_{decay} \approx 0.32$.
> * **Constraints & Failure Rate:** To prevent image distortion, our code performs strict aspect ratio similarity checks (resolution ratio restriction $< 1.2$). If a matching image isn't found in 50 attempts, augmentation is skipped. The failure rate is approx. **20%**.
>
> Therefore, the actual effective augmentation rate is:
> $$P \approx 0.32 \times (1 - 0.2) = 0.256$$
> Thus, maintaining a similar extra exposure for hard samples, the training time for Oversampling would be roughly 25.6% longer than SFT (i.e., $38.2 \times 1.256 \approx 47.98$ hours), which matches the **47.2 hours** reported in Table 3. We have clarified these calculations in the final PDF (Section D.2.1, line 1026).
>
> **3. "Effective Batch Size" Ablation (Table R5):**
> To address the concern that performance gains might come from a larger effective batch size, we conducted an experiment ("Expand_bs") where the images destined for a montage were instead placed into the same batch as independent samples (using the same training parameters).
>
> **Table R5: Effective Batch Size Ablation**
> | Method | GMAI(%) | Fundus(%) | R_lesion(%) | R_rare(%) | Average(%) |
> | :--- | :---: | :---: | :---: | :---: | :---: |
> | Vanilla SFT (Baseline) | 70.8 | 69.8 | 66.5 | 60.1 | 66.8 |
> | **MontageAug** | **73.1 (+2.3)** | **71.3 (+1.5)** | **69.3 (+2.8)** | **63.4 (+3.3)** | **69.3 (+2.5)** |
> | Expand_bs | 68.9 (-1.9) | 70.5 (+0.7) | 66.9 (+0.4) | 59.6 (-0.5) | 66.5 (-0.3) |
>
> The *Expand_bs* setting performed comparably to Baseline SFT but significantly lower than MontageAug, especially on rare categories (R_rare: 59.6% vs. 63.4%). This confirms that MontageAug's improvement stems from **learning in richer context scenarios**, rather than simply increasing throughput.
>
> ---
>
> ### Response to Q2 (Regarding Training Time)
>
> The reviewer raised a very professional question. The marginal increase in MontageAug training time in Table 3 is mainly attributed to:
>
> 1.  **Lower Effective Rate:** As detailed above, the model processes Montage images in only ~25.6% of iterations, not 40%.
> 2.  **Non-linear Growth of Tokens:** InternVL uses a "local tiles + global thumbnail" approach to represent images. For a standard **$1024 \times 1024$ fundus image**, the model generates **10 tiles in total** ($3 \times 3$ local tiles + 1 global thumbnail). For a Montage image (e.g., $1024 \times 2048$), theoretically possessing twice the pixel area, our code enforces a strict constraint of **max_dynamic_patch=12**. Consequently, the model dynamically downsamples tiles to fit this limit. Thus, visual tokens per augmented sample effectively increase by only **20%** (from 10 to 12 tiles), rather than doubling.
> 3.  **Hardware Saturation:** The total increase in visual tokens is estimated at only ~5% ( $25.6\\% \times 20\\%$ ). On NVIDIA A100 GPUs, this slight increase in FLOPs is "absorbed" by the massive parallel core architecture and does not translate linearly into wall-clock time, especially when bottlenecks include data loading and LLM inference.
>
> ---

---

> ### Author Response · Authors · 2025-11-21
>
> ### Response to W4 and W9 (Dataset Details)
>
> We thank the reviewer for the feedback. We have clarified the dataset generation process in detail in the paper and add references (line 258) to the Appendix B.1 (line 769) in the revised PDF. Our training set consists mainly of two parts:
>
> * **Main Data (approx. 97%):** We obtained authorized use of the **FundusGen** dataset [1]. This part contains about 294K VQA pairs. FundusGen is a high-quality dataset built specifically for collaborative reasoning in fundus photography, which already contains rich VQA pairs.
> * **Supplementary Data (approx. 3%):** We additionally constructed about 6K data points from the open-source training sets of **MuReD** and **RFMiD 2.0**. The dataset analyzed in Figure 1 of the original manuscript actually refers primarily to the FundusGen dataset (approx. 294K VQA pairs) for which we obtained authorization. We have explicitly stated this in the caption of Figure 1.
>
> Regarding report generation for the supplementary images, we used **Gemini 2.5 Pro** as a powerful MLLM to generate medical reports. From the results in Table 1, Gemini 2.5 Pro demonstrates strong basic capabilities in this field. The template (prompt) we used is shown below; it instructs the model to act as a professional ophthalmologist and generate objective fundus image feature descriptions based on the image itself, reliable and detailed structured labels (`labels_json`), and authoritative disease descriptions (`disease_descriptions` from the RFMiD 2.0 paper [2]).
>
> > f"You are a professional fundus imaging specialist. You have a color fundus image of a patient, along with detailed corresponding labels: {labels_json}.\n\n"
> > f"{disease_descriptions}\n\n"
> > "Combined with this information, please provide a detailed description of this fundus image. This description may reference findings that may aid in the diagnosis, but should not directly cite or infer a diagnosis. Please note that only objective descriptions of fundus images are required."
>
> *[1] Xinyao Liu and Diping Song. Constructing ophthalmic mllm for positioning-diagnosis collaboration through clinical cognitive chain reasoning. ICCV 2025.*
>
> *[2] Sachin Panchal et al. Retinal fundus multi-disease image dataset(rfmid)2.0. Data, 2023.*
>
> ---
>
> ### Response to W5 (MontageAug Text Template)
>
> We apologize for the oversight. We only provided illustrative explanations in Figure 3 of the main text and Figure 6 of the appendix but did not explicitly write out the MontageAug text template used in the main text.
>
> * **Horizontal Montage:** The connecting text template is: *"This image is a montage of K parts stitched from left to right. \nQuestion for Part k: ..."*
> * **Vertical Montage:** The connecting text template is: *"This image is a montage of K parts stitched from top to bottom. \nQuestion for Part k: ..."*
>
> ---
>
> ### Response to W7 (Oversampling Extension Experiment)
>
> We thank the reviewer for the rigorous suggestion. For a fair comparison, the oversampling frequency in the original text was set to align with the effective exposure rate of hard samples in MontageAug ($\approx 25.6\%$) to rule out the influence of exposure count. To address your concerns, we supplemented extended experiments for the Oversampling strategy, covering doubled oversampling rates and longer training epochs.
>
> **Table R6: Oversampling Extension Results**
>
> | Method | Training Time(h) | GMAI(%) | Fundus(%) | R_lesion(%) | R_rare(%) | Average(%) |
> | :--- | :---: | :---: | :---: | :---: | :---: | :---: |
> | Vanilla SFT (Baseline) | 38.2 | 70.8 | 69.8 | 66.5 | 60.1 | 66.8 |
> | Oversampling (Original) | 47.2 | 71.5 | 69.5 | 67.6 | 60.1  | 67.2 |
> | **Oversampling (Rate Doubled)** | 56.3 | 71.2 | 69.8 | 68.2 | 60.7 | 67.5 |
> | **Oversampling (3 Epochs)** | 71.0 | 71.2 | 68.2 | 67.1 | 60.1 | 66.7 |
> | **MontageAug** | 38.3 | **73.1** | **71.3** | 69.3 | **63.4** | **69.3** |
> | **MontageAug (3 Epochs)** | 57.5 | 72.4 | 71.3 | **69.6** | 63.0 | 69.1 |
>
> When we doubled the oversampling rate (meaning the model saw far more hard samples than in MontageAug), its average performance (67.5%) improved slightly but still could not surpass MontageAug (69.3%). Furthermore, increasing training epochs actually led to a performance decrease (average dropped to 66.7%). Moreover, in terms of training cost, the extended oversampling strategies brought significantly increased computational overhead. Therefore, we maintain a 2-epoch training setting for the medical task of fundus images.
>
> We would also like to clarify that we do not treat "Training Epochs" as a hyperparameter to be heavily tuned for MontageAug. Our method is designed to be "plug-and-play." As demonstrated in our **General Response** regarding the **UniMERNet** experiment in Mathematical Expression Recognition, MontageAug achieves significant improvements while strictly adhering to the original method's training schedule ($\approx 36$ epochs), without the need for specific epoch adjustments.
>
> ---

---

> ### Author Response · Authors · 2025-11-21
>
> ### Response to W11 (Clarification on Epoch Statement)
> We sincerely thank the reviewer for this detailed observation. We agree that our original conclusion regarding "continuous improvement," based on incomplete epoch data, was imprecise. To investigate this issue more deeply, we conducted an additional experiment (under the same MontageAug settings as Table 2) by training for 3 epochs. The updated experimental results are presented below:
> | MontageAug Settings | GMAI(%) | Fundus(%) | R_lesion(%) | R_rare(%) | Average(%) |
> | :--- | :---: | :---: | :---: | :---: | :---: |
> | 1 epoch | 67.0 | 66.5 | 65.5 | 60.1 | 64.8 |
> | **2 epochs (Default)** | **67.6** | **67.3** | 67.2 | **61.2** | **65.8** |
> | 3 epochs | 67.0 | 66.8 | **67.6** | **61.2** | 65.7 |
>
> As the results clearly indicate, the primary performance gains occur during the early training phase (from 1 to 2 epochs). By the 3rd epoch, the growth curve noticeably plateaus and does not exhibit continuous growth (GMAI and Fundus scores slightly decreased, while R_rare stabilized). Although we did not extend the experiment to a 4th epoch due to time and computational constraints, the current data sufficiently demonstrates that performance does not increase linearly indefinitely.
>
> Consequently, we have revised the statement in line 411 (line 420 in the revised PDF) replacing it with a description that aligns more strictly with our experimental results.
>
> ---
>
> ### Other Responses
>
> * **Response to W8 and W15 (Modification of Generalizability Claim):** We acknowledge that claiming general applicability to the entire field of medical image analysis based solely on ophthalmology data in the original manuscript was indeed not rigorous enough. We have modified the relevant wording in the revised manuscript to be more precise and objective. To address this concern and substantially extend the applicability boundary of our method, we added new experiments during the rebuttal phase (see **General Response**).
> * **Response to W10 (Supplementary Explanation of Test Set):** We stated the total number of fundus images in the test set in the main text. For more detailed category counts, we supplement as follows: The fundus images subset of GMAIMMBench includes 60 categories. Most categories (approx. 90%) contain 5 test data; of the remaining categories, about half have fewer than 5 test data, and half contain 10 test data. FundusMMBench contains 31 categories, with 20 data samples per category.
> * **Response to W12 (Explanation regarding LLaVA framework selection):** We understand the reviewer's concern. The choice of LLaVA-1.5 was not to demonstrate the best performance of the method (which has been proven by InternVL), but to explore the performance boundaries of the method. LLaVA-1.5 is currently the most standard and easily reproducible open-source baseline in the community. Our motivation was to establish a fair, reproducible benchmark to purely evaluate the effect of MontageAug itself.
> * **Response to W6, W13, and W14 (Writing Modifications):** Thank you for your detailed suggestions. We have incorporated the following revisions in the updated PDF:
>     * **Table Captions:** Clearly label the metrics used (accuracy) and the meaning of bold values. Add the reference to table 1 in the table 6 (previous table 3, line 1004).
>     * **Citations & Links:** Explicitly add the reference in line 126.
>     * **Flow:** Optimize the transition paragraph before methods comparison (line 095). Optimize the transition paragraph between baseline methods and our method (line 355).
>     * **Claim Correction:** Limit "practical value for medical image analysis" to the scope supported by our experimental evidence (line 29, line 138).

---

### Official Review · Reviewer_x45S · 2025-11-03

**Soundness:** 3
**Presentation:** 2
**Contribution:** 3
**Rating:** 6
**Confidence:** 3

**Summary:**

This paper aims to tackle a well-posed problem: instruction-tuned VLMs struggle on long‑tail categories, particularly in ophthalmic fundus imaging, where rare diseases (“tail classes”) have few labeled instances. In this problem, the training data are instruction–answer pairs and the objective is to improve downstream multiple‑choice diagnostic accuracy especially on tail categories, while preserving image–text semantic consistency. Specifically, the proposed method composes $k$ images into a grid montage and synchronously composes the paired texts with a deterministic template, yielding hard positive examples that are visually richer but semantically perfectly aligned with their new text. A hard-sample prioritization sampler biases secondary images toward (i) rare categories and (ii) samples mispredicted by a strong VLM, identified by an evaluator LLM. A training item is replaced by a montage with a predefined probability $\alpha$.

Empirically, the method is evaluated using the dataset constructed using ~200k fundus images and ~300k instruction pairs with 11/4 ratio of normal and harm examples and on four fundus benchmarks. Under equal budgets on InternVL‑Chat‑V3.0‑8B, MontageAug improves average accuracy by 2.5% percents. In contrast, oversampling matched for tail exposure gives +0.4 on average, while Mixup and RetinaLogos-based generation reduce performance. Against specialized medical VLMs and closed models (e.g., GPT-4o), the InternVL model fine-tuned with the proposed method is competitive or better on this benchmarking suite. Finally, a generality probe on LLaVA‑1.5 shows benefits on compositional reasoning but degradation on fine‑grained perception, attributed to montage downsampling and spatial‑label incompatibilities.

**Strengths:**

1. Figure 1 indeed ties head–tail skew to accuracy collapse, justifying a tail‑focused augmentation that preserves alignment.
2. Composing both vision and text enables the proposed method to avoid label noise that plagues baselines Mixup/generation.
3. The comparisons are generally fair and extensive.
4. The method does not introduce significant computing overhead, as shown by Table 3, where the proposed method shows nearly identical training time to vanilla and a lower cost than generation.

**Weaknesses:**

1. Weak matching is currently dependent on GPT-4o without prompt/post-processing details. Meanwhile, the robustness to other open evaluators is unreported.
2. The hard-sample pool depends on Qwen2.5-72B judgments. The sensitivity to thresholding and the evaluator are not analyzed.
3. The strongest results are in fundus VQA-style tasks, while the general-domain LLaVA results are relatively mixed.
4. The authors do not provide control for image-only montage or alternate templates. In this case, the relative role of textual scaffolding is not that clear.

**Questions:**

1. Can you release the GPT-4o prompts and normalizers used for weak matching and replicate with other open-sourced models to quantify the evaluator variance?
2. Can you compare full montage, image-only montage (i.e., keep original text), text-only concatenation (i.e., no visual montage), and alternate templates to isolate the effects?
3. It is also suggested to add a public long-tail benchmark outside medicine (i.e., single-image evaluation) to assess the cross-domain utility of this method.
4. For fixed-resolution encoders (e.g., LLaVA-1.5), have you attempted with tiled cropping/higher-resolution encoders/etc to recover MME performance?

---

> ### Author Response · Authors · 2025-11-21
>
> Dear Reviewer x45S,
>
> We greatly appreciate your recognition of the four main strengths of this work (#11887): the soundness of motivation, avoidance of label noise, extensiveness of experiments, and training efficiency.
>
> ### Response to W1 and Q1 (Regarding post-processing details during evaluation)
>
> Thank you for pointing this out. Our evaluation process on fundus color photography referenced the custom MCQ (multiple choice) dataset settings of **VLMEvalKit** (https://github.com/open-compass/VLMEvalKit) [1].
>
> We append *"Answer with the option's letter from the given choices directly."* to the tested question, requiring the model to reply only with the option. The post-processing flow follows: **Rule Matching $\rightarrow$ LLM Judgment**. Statistics show that less than **1%** of samples require LLM judgment.
>
> We have added the detailed evaluation settings to the appendix and include the citation for VLMEvalKit (line 948). Regarding evaluator variance: for the inference results in Table 1 of the main text, when we switched the post-processing evaluator to **Gemini 2.5 Pro** or **Qwen2.5-72B**, the final accuracy remained consistent.
>
> **The post-processing prompt is as follows:**
> ```text
> You are an AI assistant who will help me to match an answer with several options of a single-choice question.
> You are provided with a question, several options, and an answer, and you need to find which option is most similar to the answer.
> If the meaning of all options are significantly different from the answer, output Z.
> Your should output a single uppercase character in A, B, C, D (if they are valid options), and Z.
>
> Example 1:
> Question: What is the main object in image?
> Options: A. teddy bear B. rabbit C. cat D. dog
> Answer: a cute teddy bear
> Your output: A
>
> Example 2:
> Question: What is the main object in image?
> Options: A. teddy bear B. rabbit C. cat D. dog
> Answer: Spider
> Your output: Z
>
> Example 3:
> Question: {}?
> Options: {}
> Answer: {}
> Your output:
> ```
> *References:* [1] Haodong Duan et al. Vlmevalkit: An open-source toolkit for evaluating large multi-modality models. ACM Multimedia, 2024.
>
> ---
>
> ### Response to W2 (Details regarding "hard-sample pool" evaluator selection, threshold selection, and robustness analysis)
>
> We thank the reviewer for the valuable feedback on the construction of the "hard-sample pool." We used an LLM to combine the question and reference answer to score the answer provided by InternVL2.5-8B on a scale of 1-10. Samples with a score lower than 3 entered the hard sample pool.
>
> * **Reason for Threshold Selection:** First, we clarify that the threshold of 3 was not an arbitrarily chosen hyperparameter tuned to improve performance. A score of $< 3$ (1-2 points) represents serious hallucinations or errors, while $\ge 3$ represents at least partial correctness. This is a quality-based binary distinction; in this setting, sensitivity analysis on this threshold is somewhat less applicable.
> * **Reason for Using Qwen2.5-72B:** The choice of Qwen2.5-72B was driven by practical needs for processing a large-scale dataset (approx. 300k samples). Using commercial APIs (e.g., GPT-4o) to process the entire dataset would be cost-prohibitive. Qwen2.5-72B represents a state-of-the-art open-source model allowing for efficient local deployment.
> * **Consistency and Robustness Analysis:** To address the issue of dependency on a specific evaluator, we conducted a consistency analysis. We randomly sampled 500 samples for validation (using the same prompt and threshold $< 3$). The **Intersection over Union (IoU)** of the hard sample pool between Qwen2.5 and GPT-4o was **91.2%** (125/137), and the IoU with Gemini 2.5 Pro was **90.7%** (127/140). This indicates that the choice of a specific model has a minor impact on the composition of the hard sample pool. We have included this consistency analysis in the appendix (line 955) of the revised PDF.

---

> > ### Author Response · Authors · 2025-11-21
> >
> > ### Response to W3, Q3, and Q4 (Regarding general domain and resolution)
> >
> > We thank the reviewer for the keen observation. Regarding the generalizability of the MontageAug method, we have added cross-domain experimental results (Chest X-ray and Math Formula Recognition) in the **General Response**.
> >
> > We greatly appreciate the constructive suggestion in Q4 to introduce "Tiled Cropping"/"Higher Resolution" (techniques used in LLaVA-NeXT) to mitigate the resolution loss caused by Montage.
> >
> > * **Reason for not adopting:** In this paper, we used the standard setting of LLaVA-1.5 (fixed $336 \times 336$ input). The core motivation was to establish a fair, reproducible baseline to purely assess the performance boundaries of the data augmentation strategy itself.
> > * **Future Outlook:** We agree that the maximum value of MontageAug should be realized under the precondition of dynamic resolution (as shown by our success on InternVL). The experimental observations on LLaVA reveal the deep relationship that data augmentation cannot exist independently of model architecture. Our findings may provide an insight for future research: efficient multi-modal data augmentation strategies (like MontageAug) should be **Data-Model Co-designed** to fully release their potential.
> >
> > ---
> > ### Response to W4 and Q2 (Ablation study on MontageAug design)
> >
> > Thank you for your meticulous consideration. We supplemented this ablation experiment under the same settings as Table 2.
> >
> > * **MontageAug (no textual scaffolding):** No templates are added to the text end; instead, the two original text descriptions $T_{main}$ and $T_{secondary(s)}$ are simply concatenated.
> > * **MontageAug (image montage only):** Under the $k=2$ setting, randomly keep the text annotation corresponding to one image and the corresponding connecting text.
> > * **MontageAug (no meta-instruction):** Keep the block structure like "Part k", but remove the introductory sentence at the beginning (i.e., "This image is a montage of...").
> >
> > **Table R4: MontageAug Design Ablation Study Results**
> >
> > | Method | Parameters | GMAI | Fundus | R_lesion | R_rare | Average |
> > | :--- | :---: | :---: | :---: | :---: | :---: | :---: |
> > | Vanilla SFT | 2e | 66.3% | 65.3% | 65.5% | 59.6% | 64.2% |
> > | **MontageAug (Ours)** | 2e, $\alpha=0.4, k=2$ | **67.6%** | **67.3%** | **67.2%** | **61.2%** | **65.8%** |
> > | w/o textual scaffolding | 2e, $\alpha=0.4, k=2$ | 56.7% | 59.2% | 58.0% | 53.0% | 56.7% |
> > | Image montage only | 2e, $\alpha=0.4, k=2$ | 64.1% | 64.0% | 62.8% | 54.4% | 61.3% |
> > | w/o meta-instruction | 2e, $\alpha=0.4, k=2$ | 65.4% | 64.8% | 64.5% | 57.4% | 63.0% |
> >
> > The experimental results show that removing textual scaffolding (56.7%) leads to a significant drop in performance, proving that precise image-text alignment is crucial. Furthermore, "image montage only" or "no meta-instruction" both performed worse than the complete MontageAug (65.8%), confirming the necessity of the holistic design of **"Vision + Text + Structured Instruction."**
> >
> > *Note: We did not adopt a "text montage only" setting because this would destroy image-text alignment and introduce serious hallucinations, contradicting our original design intent.*

---

### Author Response · Authors · 2025-11-21
**General Response and Enhanced Cross-Domain Evaluation**

We first thank all reviewers for recognizing the core contributions of MontageAug, including: the soundness of the method and sufficiency of motivation (Reviewer x45S, WyrV, xo91), high efficiency and low overhead (Reviewer x45S), and the effectiveness of the experiments (Reviewer x45S, WyrV, xo91).

We have **updated PDF** with the revised parts marked in blue. Below are our clarifications and supplementary experiments addressing the common concern regarding the method's cross-domain generalization:

### Common Concern: Generalizability & Cross-Domain Effectiveness

**Response:**
We acknowledge that the performance of MontageAug in the general domain, as presented in the original manuscript, was indeed less impressive than its performance in the medical ophthalmology vertical domain.

We attribute this to two primary factors:
1.  **Resolution Limitations:** As noted in the paper, the fixed resolution of models like LLaVA limits the fidelity of montaged images.
2.  **Data Characteristics (Semantic & Stylistic Variance):** Unlike the standardized nature of medical imaging, general-domain datasets exhibit **significant semantic and stylistic differences** between image-text pairs. The composition of two highly heterogeneous "in-the-wild" images (e.g., a sketch and a photograph) may fail to enable effective contextual semantic transfer, consequently being unable to adequately stimulate the model's reasoning ability. This results in a mixed performance boost in the general domain. In contrast, the montage proves even more effective within a vertical domain, delivering significant and consistent performance improvements.

**Refined Scope:** Consequently, in the revised manuscript, we have refined our contribution to define MontageAug specifically as **an augmentation solution targeted at "Long-Tail Vertical Domains"** (where data is scarce but stylistic consistency is higher), rather than a universal pre-training tool.

To further verify its cross-domain utility, we have added the following experiments, which strongly support the effectiveness of MontageAug in different vertical domains.

#### 1. Medical Domain Extension: Chest X-ray (CXR)
To demonstrate that MontageAug is not limited to ophthalmology, we applied it to Chest X-rays, another high-stakes medical modality.
* **Dataset:** We conducted cross-domain experiments on a subset of the MIMIC-CXR chest X-ray dataset. The dataset is derived from the open-source **13K images and annotations** (https://physionet.org/content/mimic-cxr-jpg/2.1.0/). We utilized GPT-4o to generate medical reports based on the images and annotations.
* **Settings:** We conducted comparative experiments with different methods using **InternVL3.0-1B** instruction tuning under identical hyperparameters.
* **Evaluation:** The evaluation used the Chest X-ray subset of GMAIMMBench [1] (Data volume: 115 entries). MontageAug directly combined randomly sampled images without filtering for hard examples. The experiments were repeated three times.
* **Results:** As shown below, the model trained with MontageAug achieved a **1.7%** improvement in disease recognition compared to Vanilla SFT.

**Table R1: Accuracy of different methods on GMAIMMBench (Chest X-ray subset)**
| Method | GMAIMMBench (Chest X-ray subset) |
| :--- | :---: |
| InternVL3.0-1B (Zero-shot) | 34.8% |
| Vanilla SFT | 37.4% |
| **MontageAug, RandomSampling** | **39.1%** |

---

> ### Author Response · Authors · 2025-11-21
>
> #### 2. Non-Medical Domain Verification: Mathematical Expression Recognition (MER)
> To address the request for a non-medical benchmark, we selected Mathematical Expression Recognition (MER). This task faces severe long-tail distribution problems, such as rare complex long formulas. We re-ran the methods following the experimental settings recommended in the original papers, controlling the variable to solely be the addition of the MontageAug method.
> * **Background:** Simple short formulas are "Head" classes, while **Complex Print Expressions (CPE)** and **Handwritten Expressions (HWE)** are not only scarce but also extremely difficult to recognize ("Tail/Hard").
> * **Settings:** We selected the SOTA model **UniMERNet** [2]. Following its official configuration, we trained on the UniMER dataset (**1.06M samples**) for 300k iterations using 16 A100 GPUs. We treat CPE (**5,921 samples**) and HWE (**6,332 samples**) in the test set as "long-tail/hard" categories. MontageAug synthesizes "longer, more complex" formula samples by randomly splicing two formula images while maintaining precise semantic alignment.
>
> **Experimental Results A: UniMER Validation Set Performance.**
> As shown in Table R2, MontageAug consistently outperforms the baseline. Crucially, the gains are most significant on the long-tail/hard subsets (CPE and HWE), verifying MontageAug's ability to improve robustness by synthesizing high-complexity samples.
>
> **Table R2: Performance Comparison on UniMER Validation Set (Metrics: BLEU ⬆, Edit Distance ⬇)**
> | Method | Simple Printed (SPE) | Complex Print (CPE) [hard] | Screen-Captured (SCE) | Handwritten (HWE) [hard] |
> | :--- | :---: | :---: | :---: | :---: |
> | | bleu ⬆ / edit ⬇ | bleu ⬆ / edit ⬇ | bleu ⬆ / edit ⬇ | bleu ⬆ / edit ⬇ |
> | UniMERNet-Small | 0.911 / 0.064 | 0.917 / 0.065 | 0.569 / 0.243 | 0.891 / 0.075 |
> | **+ MontageAug** | 0.915 / 0.061 | **0.934 (+1.7%)** / 0.053 | 0.577 / 0.241 | **0.904 (+1.3%)** / 0.067 |
>
> **Experimental Results B: Public Benchmarks (CROHME & HME100K).**
> To meet the requirement regarding public benchmarks, we evaluated on the recognized **CROHME 2014 (986 samples), 2016 (1,147 samples), 2019 (1,199 samples) [3-5], and HME100K Test (24,607 samples) [6]** datasets. MontageAug achieved new SOTA results by significantly improving Expression Accuracy (ExpRate).
>
> **Table R3: Performance Comparison on Public Handwritten Datasets (Metric: ExpRate ⬆)**
> | Method | CROHME 2014 | CROHME 2016 | CROHME 2019 | HME100K |
> | :--- | :---: | :---: | :---: | :---: |
> | Baseline | 59.43% | 57.63% | 55.88% | 65.08% |
> | **+ MontageAug** | **62.17% (+2.7%)** | **61.81% (+4.2%)** | **58.63% (+2.8%)** | **66.57% (+1.5%)** |
>
> **References:**
>
> [1] Jin Ye et al. Gmai-mmbench: A comprehensive multimodal evaluation benchmark towards general medical ai. NeurIPS 2024.
>
> [2] Bin Wang et al. Unimernet: A universal network for real-world mathematical expression recognition. arXiv 2024.
>
> [3-5] Mouchere et al. ICFHR/ICDAR CROHME Competitions (2014, 2016, 2019).
>
> [6] Yuan et al. Syntax-aware network for handwritten mathematical expression recognition. CVPR 2022.

---

### Author Response · Authors · 2025-12-02
**Paper Overview and Discussion Summary**

Dear PCs, ACs, and Reviewers,

Thank you very much for your valuable contributions. To assist the ACs in their assessment, we provide below a summary of the key points.

## 1. Motivation & Paper Overview
- Motivation: Real-world vertical domains (e.g., medical imaging) are plagued by long-tail distributions, where rare but critical categories ("tail classes") are scarce. Standard Vision-Language Model (VLM) training often overfits to "head" classes or learns biased shortcuts.
- Method Overview: To address this, we propose a compositional data augmentation(**MontageAug**) that creates "hard positive samples" by stitching images (prioritizing tail classes) into a montage and synchronously generating a structured, semantically consistent text.
- Efficiency & Results: MontageAug achieves the best accuracy on multiple fundus image benchmarks, outperforming Vanilla SFT, Oversampling, and Mixup, particularly on rare disease recognition. With the added cross-domain evidence (Chest X-ray and Math OCR) , we have demonstrated its efficacy extends beyond ophthalmology. Notably, the method is designed for simplicity and efficiency; it requires a minor increase in training overhead compared to Vanilla SFT. This presents a distinct efficiency advantage over methods like Oversampling or Generation, which introduce significant computational costs.
### Summary of Strengths
We appreciate the reviewers' recognition of the specific strengths of this work:
- **Sound Motivation & Design**: The work is well motivated in the need of improving models for long-tailed problems (Reviewer x45S: Strength 1, Reviewer WyrV: Strength 1). Reviewers agreed it is sound to prioritize semantic consistency in VLM augmentation, noting that our compositional design successfully avoids the label noise that plagues baselines like Mixup (Reviewer xo91: Strength 1,2 , Reviewer x45S: Strength 2).
- **Simplicity & Generalizability**: The method convinces with its simplicity and broad applicability to most VLMs and tasks. (Reviewer WyrV: Strength 1)
- **Performance & Fairness**: The experimental comparisons are recognized as fair and extensive (Reviewer x45S: Strength 3). The results demonstrate that MontageAug surpasses traditional baselines, including simple oversampling and single-modality augmentation (Reviewer xo91: Strength 2).
- **High Efficiency**: A key practical advantage is that the method does not introduce significant computing overhead—showing lower costs than generation-based methods (Reviewer x45S: Strength 4).

---

> ### Author Response · Authors · 2025-12-02
>
> ## 2. Key Rebuttal Highlights & Additional Experiments
>
> During the rebuttal phase, we comprehensively responded the concerns raised by the reviewers through new experiments and clarifications. The following is a summary of the key points to representative questions.
>
> ### A. Generalizability & Scope Refinement (Reviewer x45S, xo91)
>
> Reviewers noted that the initial claim of "general domain" effectiveness was over-broad given the mixed results on LLaVA (Appendix E).
> - **Refinement**: We have refined our contribution scope to **"Long-Tail Vertical Domains"** (e.g., medical, specialized OCR), where data is scarce but context consistency is higher.
> - **New Evidence 1 (Medical Extension)**: We applied MontageAug to **Chest X-rays** (MIMIC-CXR subset). Results show a 1.7% accuracy improvement over Vanilla SFT.
> - **New Evidence 2 (Non-Medical Extension)**: We validated MontageAug on **Mathematical Expression Recognition (MER)** using the SOTA model UniMERNet. MontageAug achieved significant gains on "hard/tail" subsets (BLEU: +1.7%, +1.3%) and achieved promising results on public benchmarks (+2.7% ExpRate) using default hyperparameters (plug-and-play) without specific tuning.
>
> ### B. Mechanism Validation: Context or Throughput (Reviewer WyrV)
>
> Reviewer WyrV questioned whether performance gains stemmed from the montage context or simply an increased **"effective batch size"** (throughput).
> - **New ablation**: We conducted an *Expand_bs* ablation experiment where images destined for a montage were instead added to the batch as independent samples.
> - Results: Simply increasing throughput resulted in lower performance (-2.8% avg) compared to MontageAug, specifically failing on rare classes. This confirms that the **visual composition context** is the driver of improvement, not larger batch size.
>
> ### C. Component Necessity: Necessity of Textual Scaffolding (Reviewer x45S)
>
> Reviewers x45S asked to isolate the effect of the textual template.
> - **New ablation**: We tested MontageAug *without textual scaffolding* (simple concatenation). Accuracy dropped significantly to 56.7% (vs. 65.8% with scaffolding). Additionally, we demonstrated that *Image Montage Only* (61.3%) and *removing the meta-instruction* (63.0%) both failed to match the full MontageAug performance (65.8%).
> - This further validates that the holistic design of *Vision + Text + Structured Instruction* is critical for success.
>
> ### D. Robustness & Model Choice (Reviewers x45S, xo91)
>
> - **Evaluator Consistency**: We addressed concerns about dependence on specific LLMs for hard sample mining; we demonstrated >90% IoU consistency between Qwen2.5, GPT-4o, and Gemini 2.5 Pro.
> - **Base Model Choice**: We justified the choice of InternVL over domain-specific LLaVA-Med by showing InternVL achieves higher performance (66.8% vs 56.0%) and better supports dynamic inputs necessary for effective high-resolution MontageAug.
>
> ### E. Revisions & Presentation
>
> - We have updated the PDF (**highlighted in blue**) to address writing and presentation issues, including clarifying table captions, refining the scope of claims, and optimizing transitions.

---

### Meta-Review · Area_Chair_KLXx · 2026-01-10

**Summary:**

All three reviewers agree that the paper introduces a simple augmentation strategy for long-tailed tasks. Gains are clearest in ophthalmology VQA. But major concerns remain regarding experimental clarity and methodological soundness. Reviewers also question the scope of the claims.

**Reviewer x45S** liked the motivation and medical improvements but noted several gaps. These include a reliance on proprietary models for selection and missing ablations to separate visual and textual effects. **Reviewer WyrV** was more critical of the experimental design. They highlighted unclear epoch definitions and possible confounding from effective batch sizes. **Reviewer xo91** argued that the paper overstates its generality. They found the exposition confusing and noted that implementation details were missing.

In the rebuttal, the authors provided GPT-4o prompts and VLMEvalKit details. They showed the hard-sample pool is stable across different evaluators. The authors also narrowed the scope to vertical domains and added chest X-ray tests. These changes are helpful, but core problems persist. No ablations were provided to isolate the montage effects. Confounding from data exposure and batch size remains a risk. Evidence for robustness outside the medical setting is still limited.

**Reviewer Concerns:**

**Reviewer x45S** focuses on three main issues: dependence on closed-source models, missing ablations, and limited generalization. Authors released detailed GPT-4o prompts and clarified the use of VLMEvalKit. They also showed that the hard-sample pool is consistent when switching to Qwen2.5. These updates address some concerns, but the requested ablations comparing image and text effects are still missing. As a result, the source of the gains is not clearly disentangled.

**Reviewer WyrV** is primarily concerned with experimental soundness. Their comments center on epoch definitions and effective batch sizes. The rebuttal fills in some details regarding Qwen2.5-72B and the quality threshold of 3. But the key experimental concern is not resolved. Authors did not run the requested ablation with a larger batch size for the baseline. So it remains possible that improvements come from increased data exposure rather than the montage design.

**Reviewer xo91** raised points about over-claimed generalizability and missing methodological details. In response, the authors reframed the work to target vertical domains rather than universal augmentation. They added experiments on chest X-ray and math formulas to show cross-domain applicability. This softens the over-claim. However, the paper still lacks comparisons to medical VLMs like LLaVA-Med. These gaps leave the reviewer’s skepticism intact.

**Reviewer Scores:**

**Reviewer x45S (Original: 6 → Predicted: 6)**
This reviewer is the most positive and finds the method empirically useful. The rebuttal provided detailed prompts and showed stability across different evaluators. These additions likely reinforce their favorable view. But the lack of ablations to separate visual and textual contributions remains a problem. I expect their score to stay at weak accept.

**Reviewer WyrV (Original: 4 → Predicted: 4)**
**Reviewer WyrV** has strong doubts about experimental soundness and clarity. The rebuttal clarifies some implementation details but does not provide the requested batch size ablation. So the concern that gains come from more data exposure is still active. These issues are central to their assessment and likely prevent a score increase.

**Reviewer xo91 (Original: 4 → Predicted: 4)**
**Reviewer xo91** is concerned about over-claiming and missing methodological details. Reframing the work as targeting vertical domains helps slightly. But the paper still lacks comparisons to specialized models like LLaVA-Med. Their skepticism about the contribution will likely lead to a recommendation to reject.

---

### Decision · Program_Chairs · 2026-01-26

Reject